# Cryo-EM reveals an extrahelical allosteric binding site at the $M_5$ mAChR

Wessel A. C. Burger [1,2,7], Jesse I. Mobbs [1,2,7], Bhavika Rana[1,2,7], Jinan Wang[3,7], Keya Joshi [3], Patrick R. Gentry[1], Mahmuda Yeasmin[1,2], Hariprasad Venugopal [4], Aaron M. Bender[5], Craig W. Lindsley[5], Yinglong Miao [3] ✉, Arthur Christopoulos [1,2,6] ✉, Celine Valant [1,2] ✉ & David M. Thal [1,2] ✉

The $M_5$ muscarinic acetylcholine receptor ($M_5$ mAChR) represents a promising therapeutic target for neurological disorders. However, the high conservation of its orthosteric binding site poses significant challenges for drug development. While selective positive allosteric modulators (PAMs) offer a potential solution, a structural understanding of the $M_5$ mAChR and its allosteric binding sites remains limited. Here, we present a 2.8 Å cryo-electron microscopy structure of the $M_5$ mAChR complexed with heterotrimeric $G_q$ protein and the agonist iperoxo, completing the active-state structural characterization of the mAChR family. To identify the binding site of $M_5$-selective PAMs, we implement an integrated approach combining mutagenesis, pharmacological assays, structural biology, and molecular dynamics simulations. Our mutagenesis studies reveal that selective $M_5$ PAMs bind outside previously characterized $M_5$ mAChR allosteric sites. Subsequently, we obtain a 2.1 Å structure of $M_5$ mAChR co-bound with acetylcholine and the selective PAM VU6007678, revealing an allosteric pocket at the extrahelical interface between transmembrane domains 3 and 4 that is confirmed through mutagenesis and simulations. These findings demonstrate the diverse mechanisms of allosteric regulation in mAChRs and highlight the value of integrating pharmacological and structural approaches to identify allosteric binding sites.

The $M_5$ muscarinic acetylcholine receptor (mAChR) belongs to the class A G protein-coupled receptor (GPCR) family. As one of five mAChR subtypes ($M_1$-$M_5$), it responds to the endogenous neurotransmitter acetylcholine (ACh)[1]. Despite its low expression levels in the central nervous system (CNS), the $M_5$ mAChR plays crucial roles, primarily localizing to dopamine-containing neurons in the substantia

pars nigra and ventral tegmental areas[2–4]. Historically, drug discovery efforts targeting the $M_5$ mAChR concentrated on antagonists and negative allosteric modulators (NAMs). This focus stemmed from evidence that $M_5$ mAChR inactivation can modulate dopaminergic signalling in the CNS[5,6], suggesting therapeutic potential for substance addiction, depression, and anxiety[7–15]. However, $M_5$ mAChR activation

[1]Drug Discovery Biology, Monash Institute of Pharmaceutical Sciences, Monash University, Parkville, Vic, Australia. [2]Australian Research Council Centre for Cryo-Electron Microscopy of Membrane Proteins, Monash Institute of Pharmaceutical Sciences, Monash University, Parkville, Vic, Australia. [3]Computational Medicine Program and Department of Pharmacology, University of North Carolina, Chapel Hill, NC, USA. [4]Ramaciotti Centre for Cryo-Electron Microscopy, Monash University, Clayton, Vic, Australia. [5]Department of Pharmacology, Warren Center for Neuroscience Drug Discovery and Department of Chemistry, Vanderbilt University, Nashville, TN, USA. [6]Neuromedicines Discovery Centre, Monash University, Parkville, Vic, Australia. [7]These authors contributed equally: Wessel A. C. Burger, Jesse I. Mobbs, Bhavika Rana, Jinan Wang. ✉e-mail: Yinglong_Miao@med.unc.edu; arthur.christopoulos@monash.edu; celine.valant@monash.edu; david.thal@monash.edu

may offer equally promising therapeutic applications. Studies using $M_5$ mAChR knockout mice revealed that this receptor mediates CNS vasculature dilation, thereby regulating cerebral blood flow[16,17]. This finding suggests that selective $M_5$ mAChR activation could benefit conditions such as Alzheimer's disease, schizophrenia, and ischaemic stroke by enhancing CNS circulation and blood flow.

The development of selective $M_5$ mAChR activators faces significant challenges, primarily due to the highly conserved orthosteric acetylcholine binding site shared across all five mAChR subtypes[18,19]. This conservation has historically hindered the development of subtype-selective orthosteric ligands. To overcome this limitation, researchers shifted focus to allosteric modulators, which target spatially distinct binding sites[20]. This strategy proved promising, with several selective allosteric modulators successfully developed for the $M_5$ mAChR[21–26]. The selectivity of these modulators likely results from non-conserved residues present in the allosteric binding sites, distinguishing them from the highly conserved orthosteric site.

$M_5$-selective positive allosteric modulators (PAMs) were initially discovered and developed based on a bromo isatin scaffold, as exemplified by the PAM VU0119498[27]. VU0119498 enhanced ACh signalling in a calcium assay at all $G\alpha_q$-coupled mAChRs ($M_1$, $M_3$, $M_5$) and was therefore chosen as a tool compound for further $M_5$ mAChR PAM optimization. A range of $M_5$ mAChR PAMs were developed based on VU0119498, and these all show $M_5$ mAChR selectivity over all mAChR subtypes with varying levels of potency and PAM activity[21,22,24]. Unfortunately, all $M_5$ mAChR selective PAMs based on the isatin scaffold exhibit a poor drug metabolism and pharmacokinetic (DMPK) profile that was hard to overcome due to the intractability of the isatin scaffold in medicinal chemistry optimization[1,24].

To overcome these limitations, high-throughput screening identified a scaffold amenable to modification[25]. This effort led to the development of 1-((1H-indazol-5-yl)sulfoneyl)-N-ethyl-N-(2-(trifluoromethyl)benzyl)piperidine-4-carboxamide (ML380), which emerged as the most potent $M_5$-selective PAM at the time[25]. ML380's ability to penetrate the CNS made it valuable for studying molecular mechanisms of allosteric modulation and as a template for further derivatives[26,28,29]. However, poor partition coefficients ultimately limited its in vivo utility[25]. Despite these incremental advances in developing $M_5$-selective PAMs, their precise allosteric binding site remained unknown, significantly limiting rational, structure-based drug design. The $M_5$ mAChR contains at least three known allosteric sites: the extracellular vestibule (ECV), the amiodarone-binding site[30], and the extrahelical EH4 pocket recognized by the $M_5$-selective NAM ML375[31]. However, none of these sites were structurally confirmed as the binding location for $M_5$-selective PAMs. Understanding the location of the $M_5$-selective PAM binding site will accelerate the development of improved compounds for in vivo use[32]. Therefore, we initiated studies to determine the binding site of $M_5$-selective PAMs, beginning with ML380.

Here, we determine the binding site of $M_5$-selective PAMs through an integrated approach combining mutagenesis, pharmacological assays, structural biology, and molecular dynamics simulations. We present cryo-EM structures of the active $M_5$ mAChR and reveal an allosteric binding site at the extrahelical interface between transmembrane domains 3 and 4 that binds selective PAMs. Our findings reveal the structural basis for $M_5$-selective allosteric modulation, providing a foundation for rational, structure-based drug design targeting this therapeutically important receptor.

## Results

### ML380 does not bind to known $M_5$ mAChR allosteric binding sites

To identify ML380's binding site, we first investigated its interaction with known allosteric sites through functional inositol monophosphate (IP1) accumulation assays. We tested ML380's activity on both

wild-type (WT) $M_5$ mAChR and mutants targeting two known allosteric sites: (1) the ECV allosteric site[33,34], using alanine mutants, and (2) the EH4 pocket used by the $M_5$-selective NAM ML375[31], using mutations that convert non-conserved residues to their $M_2$ mAChR equivalents (A113$^{3.35}$V, G152$^{4.47}$A, and L156$^{4.51}$V; superscript refers to the Ballesteros and Weinstein scheme for conserved class A GPCR residues[35]). These experiments measured three key parameters: ML380's affinity ($pK_B$), its efficacy in the system (log $\tau$), and its functional cooperativity with ACh (log $\alpha\beta$)[36]. Consistent with previous studies, ML380 demonstrated both agonism and positive cooperativity with ACh at the WT $M_5$ mAChR[28,29] and thus acts as an agonist-PAM (Ago-PAM, Fig. 1a). The EH4 pocket mutant maintained ML380 function with no significant change in affinity compared to WT (Fig. 1a, c, d, Supplementary Table 1). While some ECV alanine mutants showed significant differences in ML380's affinity, efficacy, and cooperativity parameters, the compound retained its ability to bind, activate, and modulate receptor function in all cases (Fig. 1b, c–e, Supplementary Fig. 1, Supplementary Table 1). No significant differences in receptor expression were observed at these mutants[31]. These results indicated that ML380 binds neither to the ECV allosteric site nor to the EH4 pocket used by ML375, leading us to use cryo-electron microscopy (cryo-EM) to identify its binding site.

### Structure determination of the $M_5$ mAChR in an active state

Active state structures of the $M_1$-$M_4$ mAChRs have been determined by cryo-EM[34,37,38], leaving the $M_5$ mAChR as the last remaining subtype. To determine an active state $M_5$ mAChR structure, we engineered a modified receptor construct with several key alterations: removal of intracellular loop 3 (ICL3) residues 237-421, addition of an N-terminal HA signal sequence, and incorporation of an anti-Flag epitope tag. This modified receptor was then fused to mini-$G\alpha_{sqiN}$ (hereafter referred to as $mG\alpha_q$), resulting in a chimeric construct[39,40]. The receptor-G protein complex was prepared through a multi-step process including detergent solubilization for purification, stabilization with scFv16[41], addition of apyrase to hydrolyze GDP, and inclusion of 10 μM iperoxo. Prior to cryo-EM analysis, we incubated the iperoxo-bound $M_5$ mAChR-$mG\alpha_q$ complex with 10 μM ML380 overnight on ice, followed by freezing and imaging using single-particle cryo-transmission electron microscopy (TEM) on a Titan Krios microscope. The resulting structure achieved a global resolution of 2.8 Å, providing sufficient EM density maps to position the receptor and $mG\alpha_q\beta_1\gamma_2$ complex (Fig. 1f, Supplementary Fig. 2-3, Supplementary Table 2). However, due to poor density, the scFv16 component was excluded from subsequent data analysis and modelling.

### Analysis of the active state $M_5$ mAChR

The iperoxo-bound $M_5$ mAChR-$mG\alpha_q$ complex revealed iperoxo bound to the canonical mAChR orthosteric binding site, which is characterized by several key aromatic residues (Fig. 1g, h). The rotamer toggle switch residue W455$^{6.48}$ forms the binding site's floor, while three tyrosine residues (Y111$^{3.33}$, Y458$^{6.51}$, and Y481$^{7.39}$) move inward during activation to create a tyrosine lid, separating the orthosteric site from the ECV. Consistent with previous $M_1$-$M_4$ mAChR structures[33,34,37,38], iperoxo was positioned between these aromatic residues (Fig. 1h). Additional binding site residues include the aromatic residues W162$^{4.57}$ and Y485$^{7.43}$, along with non-aromatic residues D110$^{3.32}$, N115$^{3.37}$, T194$^{5.39}$, T197$^{5.42}$, A201$^{5.46}$, N459$^{6.52}$, L188$^{ECL2}$, and C484$^{7.42}$. Two residues play crucial roles in iperoxo recognition. Residue N459$^{6.52}$ forms hydrogen bonds with iperoxo, while D110$^{3.32}$ engages in a charge-charge interaction with iperoxo's quaternary nitrogen.

The active state $M_5$ mAChR displays all the hallmarks of an active state class A GPCR. Relative to the inactive state, tiotropium bound $M_5$ mAChR, there is an 8.1 Å outward movement of transmembrane 6 (TM6) and 4.4 Å inward movement of TM7 (Supplementary Fig. 4). At the extracellular face, there is an outward movement of TM5 and

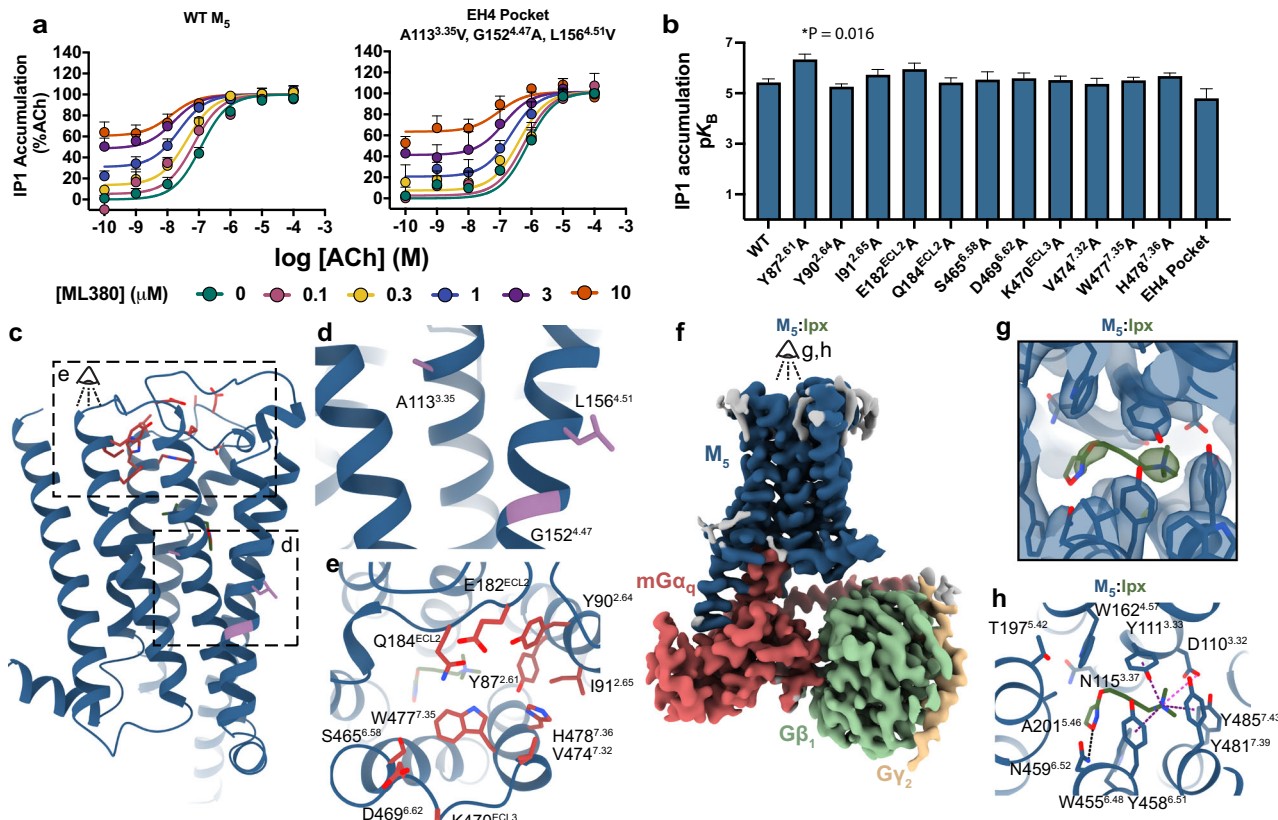

**Fig. 1 | Structural and functional analysis of ligand-receptor interactions at the $M_5$ mAChR. a** ML380-ACh interaction in IP1 accumulation assay at WT $M_5$ mAChR (left) or $M_5$ EH4 pocket mutant (right) in CHO cells. Data points are mean ± SEM values from 7 (WT $M_5$ mAChR) and 3 ($M_5$ EH4 mutant) experiments performed in duplicate. An operational model of allosterism was fit to the grouped data. **b** Effects of the $M_5$ mAChR mutations on the $pK_B$ of ML380. Data shown are the mean ± SEM of the affinity estimates derived from a global least-squares fit of an allosteric model to three independent experiments for all conditions, except for WT $M_5$ mAChR ($n = 7$). Pharmacology parameters were shared across experiments to yield a single best estimate of each parameter and its associated standard error, as derived from the nonlinear regression algorithm. This global pooled analysis approach ensured model convergence in all instances. *, significantly different from WT, $p < 0.05$, one-way ANOVA, Dunnett's post hoc test. Parameters obtained are listed in Supplementary Table 1. **c** $M_5$ mAChR mutated residues shown on receptor structure. **d** EH4 pocket residues are shown as purple sticks. **e** ECV residues are shown as red sticks. **f** Consensus cryo-EM map of $M_5$ mAChR-$mG\alpha_q$/$G\beta_1\gamma_2$ complex with iperoxo at 2.8 Å (FSC 0.143). Model colouring: dark blue (receptor), red ($mG\alpha_q$), green ($G\beta_1$), yellow ($G\gamma_1$). **g** Cryo-EM density for iperoxo in orthosteric site (local refined map, contour level 0.36). **h** Iperoxo-orthosteric site interactions. Interaction colouring: pink (charge-charge); black (hydrogen bonds); purple (cation-π). Source data are provided as a Source Data file.

inward movement of TM6 and extracellular loop 3 (ECL3) that lead to a contraction of the ECV (Supplementary Fig. 4b). These global changes in TM and ECL movements are mediated by changes in the conserved class A activation motifs including the $D^{3.49}R^{3.50}Y^{3.51}$, $P^{5.50}V^{3.40}F^{6.44}$, and $N^{7.49}P^{7.50}xxY^{7.53}$ motifs (Supplementary Fig. 4d)[42].

Our active iperoxo-bound $M_5$ mAChR structure enables the comprehensive analysis of the entire mAChR family in its active state. Comparing this $M_5$ mAChR structure with other iperoxo-bound mAChR structures reveals remarkable similarity, with root mean squared deviation (RMSD) values of 0.49−0.77 Å (Fig. 2a)[33,34,37,38]. The TM domains, ECL domains, orthosteric site residues, and iperoxo positioning are nearly identical across all mAChR subtypes (Fig. 2b−d). At the receptor-G protein interface, $G\alpha_q$- and $G\alpha_{i/o}$-coupled mAChRs show distinct α5 helix insertion angles into the TM bundle (Fig. 2f, g). In $G\alpha_q$-coupled receptors ($M_1$, $M_3$, and $M_5$ mAChRs), the α5 helix top rotates toward TM6, while in $G\alpha_{i/o}$-coupled receptors ($M_2$ and $M_4$ mAChRs), it rotates toward TM2. This pattern, however, does not extend to the G protein N-terminus (Fig. 2e), possibly due to the varying use of scFv16, Nb35, and N-terminal chimeras across different structures. Additionally, the $G\alpha_q$-coupled $M_1$, $M_3$, and $M_5$ mAChR structures display a more extended and resolved TM5.

In line with our mutagenesis results (Fig. 1a, b), no cryo-EM density was observed for ML380 in the ECV allosteric site or in the EH4 binding

pocket (Fig. 3a, b). Following focused refinement with a mask, some cryo-EM density was observed parallel to TM1 and TM7 in the ECV (coloured green Fig. 3c); however, this density was commonly observed in other mAChR structures and is likely a lipid molecule[43]. We also observed partial density directly below the EH4 pocket at the bottom of the TM2,3,4 interface (coloured orange in Fig. 3b, c). To investigate whether this partial density reflects ML380 occupancy at its allosteric binding site, we conducted radioligand binding experiments using $M_5$/$M_2$ TM chimera mutants[31]. We measured interactions between ACh and [³H]-*N*-methyl scopolamine ([³H]-NMS) with increasing ML380 concentrations. If ML380 binds at the TM2,3,4 interface, we would expect complete loss of its allosteric modulation in both $M_5$/$M_2$ TM2,3,4 and $M_5$/$M_2$ TM3,4,5 chimeras, given ML380's selectivity profile[25]. Indeed, swapping TM2-5 completely abolished ML380's ability to modulate ACh binding (Fig. 3d, Supplementary Table 3). The $M_5$/$M_2$ TM1,7,h8 chimera showed increased ML380 cooperativity, confirming that the density parallel to TM1 and TM7 is not ML380, while suggesting that exchanging these TMs affects the receptor's global activation dynamics (Fig. 3d, Supplementary Table 3).

Although the TMs 2-5 mutagenesis results were promising, the density at the TM2,3,4 interface coincides with a common cholesterol binding site in class A GPCRs[44,45], suggesting it might represent cholesterol or another lipid. For further analysis, we performed all-atom

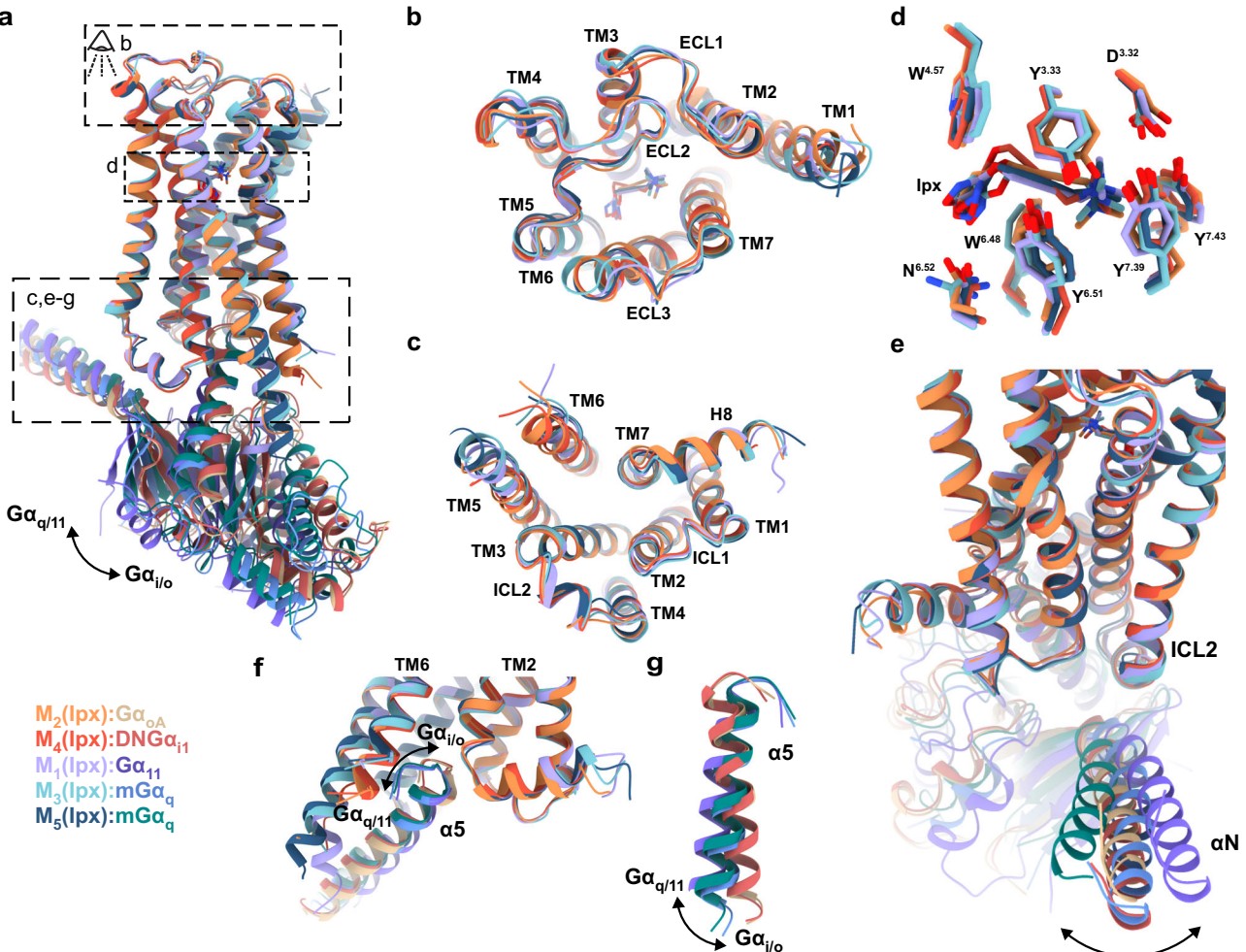

**Fig. 2 | Structural comparison of iperoxo-bound, active state M₁-M₅ mAChR structures determined by cryo-EM. a** Overall view of the M₁ to M₅ mAChRs complexed to Gα and bound to iperoxo. Model colouring: purple (M₁; PDB: 6OIJ), orange (M₂; PDB: 6OIK), cyan (M₃; PDB: 8E9Z), red (M₄; PDB: 7TRK), dark blue (M₅; PDB: 9EK0). **b** Extracellular view comparing ECLs and TM regions. **c** Intracellular view (G protein removed) comparing ICLs and TM regions. **d** Overlay of iperoxo and orthosteric binding site residues. **e** Side view comparing the αN movement of Gα. **f** Intracellular view comparing the α5 insertion of Gα. **g** Intracellular view comparing the α5 rotation of Gα. Changes are indicated by arrows.

Gaussian accelerated Molecular Dynamics (GaMD) simulations on the modelled structure of the iperoxo-M₅ mAChR-mGα_q complex with ML380 bound at the TM2,3,4 interface (Fig. 3e, f). While iperoxo maintained its cryo-EM conformation with RMSD values mostly below 2 Å (Fig. 3e), ML380 showed significant fluctuations with RMSD values of ~3–8 Å compared to the initial cryo-EM pose (Fig. 3f). The combination of mutagenesis data, GaMD simulations, and ambiguous cryo-EM density did not provide convincing support for assigning ML380 to this putative allosteric binding site.

### Use of an improved PAM, VU6007678, for structure determination

To discover the allosteric binding site for selective PAMs at the M₅ mAChR, we investigated VU6007678[26], an optimized derivative of ML380's indanyl core. VU6007678 demonstrated enhanced M₅ mAChR affinity, better positive cooperativity with ACh compared to ML380[29], and significantly improved GPCR-G protein complex stabilization. To increase our chances of obtaining a PAM-bound complex, we implemented several biochemical and pharmacological modifications. We supplemented Nb35 with scFv16 during purification for enhanced complex stability, used the endogenous agonist ACh instead of iperoxo, and maintained VU6007678 at 10 μM throughout purification rather than adding it before grid freezing as done with ML380.

These modifications improved protein purification efficiency, yielding a sample at 18 mg/mL. When applied to Au grids, the sample produced a complex resolved to 2.1 Å from 418,794 particles (Fig. 4a, Supplementary Fig. 2-3).

The high-quality cryo-EM density maps enabled precise placement of the receptor, mGα_q, β₁γ₂, Nb35, and scFv16, with clear side-chain orientations for most amino acids. The ACh-VU6007678 structure closely resembles the iperoxo structure, with an RMSD value of 0.49 Å (Supplementary Fig. 5a–d). The orthosteric binding pocket shows well-resolved density for ACh positioned beneath the tyrosine lid residues and above W455⁶·⁴⁸ (Fig. 4b, c). While ACh engages the same orthosteric binding pocket residues as iperoxo with similar orientations, W455⁶·⁴⁸ adopts a more horizontal and planar orientation with ACh-bound, similar to observations in the M₄ mAChR with ACh and iperoxo[34] (Fig. 4e). ACh forms key interactions within the orthosteric binding pocket through its quaternary ammonium group and acetyl moiety (Fig. 4f, g). The positively charged quaternary ammonium establishes a cation-π interaction with the aromatic cage formed by Y111³·³³, W455⁶·⁴⁸, Y458⁶·⁵¹, and Y481⁷·³⁹, while also engaging in a charge-charge interaction with D110³·³². The acetyl group of ACh forms a hydrogen bond with a water molecule that coordinates with N459⁶·⁵² (Fig. 4g), altogether anchoring ACh in an orientation that promotes receptor activation.

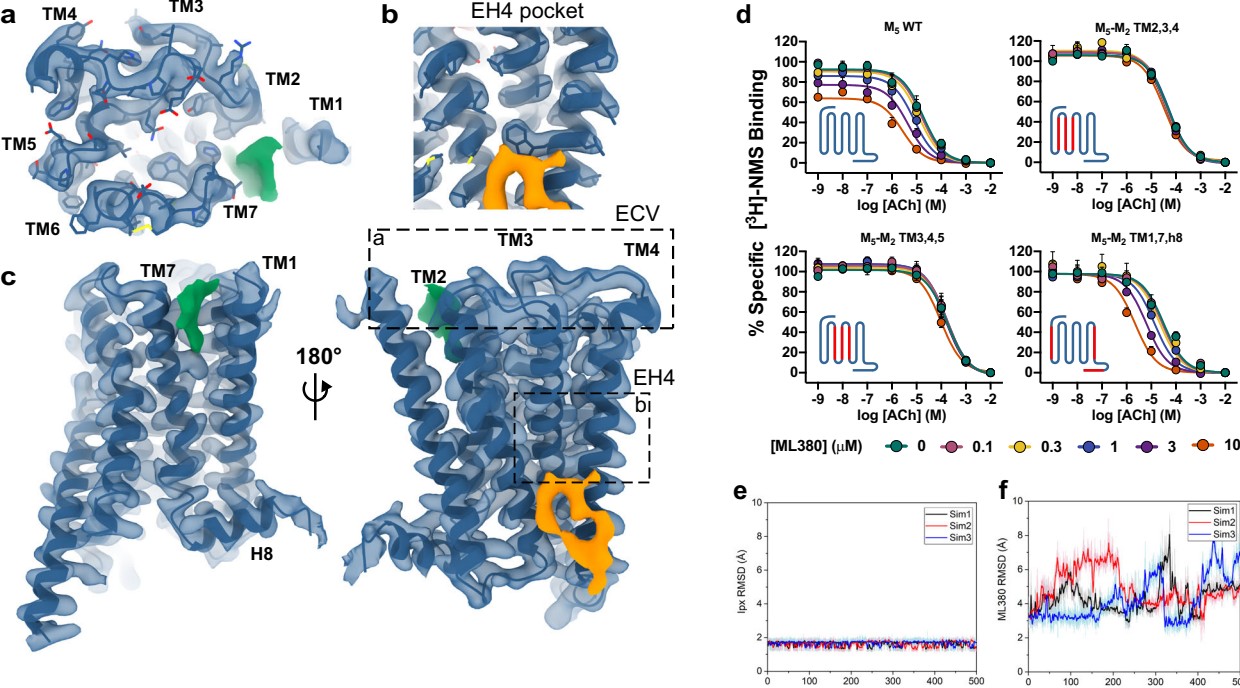

**Fig. 3 | Potential cryo-EM density for ML380. a–c** Local refined $M_5$ mAChR cryo-EM map (contour level 0.3). No ML380 density was observed in **a** ECV or **b** EH4 pocket. **c** Potential ML380 density observed parallel to TM1/TM7 (green) and the bottom of the TM2,4 interface (orange). **d** [³H]-NMS equilibrium binding studies between [³H]-NMS, ACh and ML380 at WT $M_5$ mAChR and $M_5$-$M_2$ TM chimeras. Insets show receptor cartoons (blue: $M_5$ domains; red: $M_2$ domains). Data points are

mean ± SEM values of three duplicate experiments. An allosteric ternary complex model was fit to the data. Parameters obtained are listed in Supplementary Table 3. RMSDs (Å) of **e** iperoxo in orthosteric pocket and **f** ML380 at the potential allosteric site at the bottom of the TM2,3,4 interface from GaMD simulations of the cryo-EM structure. Three simulation replicates are shown in different colours. Source data are provided as a Source Data file.

Clear, well-defined density unambiguously corresponding to VU6007678 was detected at the intracellular interface of TM3,4 and above ICL2 (Fig. 4a, b, d). The extended binding site accommodates VU6007678 through multiple interactions with the $M_5$ mAChR (Fig. 4i, k). The indanyl core, positioned at the top of the allosteric binding site, forms hydrophobic interactions with $M150^{4.45}$, $F126^{3.48}$, and $V123^{3.45}$, while engaging in an edge-to-face π-interaction with $F130^{3.52}$. The propyl chain extends toward TM4, forming a hydrophobic interaction with $I149^{4.44}$. Hydrogen bonding occurs between the carboxamide with $R146^{4.41}$ and between the sulfonyl and $K141^{ICL2}$. The indazole of VU6007678 forms a π-π interaction with $F130^{3.52}$, a cation-π interaction with $R134^{3.56}$ and at the bottom of TM3, and a hydrophobic interaction with $Y129^{3.51}$. Other residues that make up the VU6007675 binding site include $Y68^{2.42}$ and $Y138^{ICL2}$. These extensive interactions likely explain why VU6007678 has high cooperativity with ACh and its high affinity for the $M_5$ mAChR active state. Given the low affinity of VU6007678 for the inactive state $M_5$ mAChR, as observed in radioligand binding with antagonist [³H]-NMS[29], we hypothesized that these binding site residues undergo substantial rearrangement during activation. Superimposition of the ACh-VU6007678 structure with the inactive tiotropium-bound $M_5$ mAChR crystal structure[18] revealed significant conformational changes in the allosteric binding site. ICL2 remains ordered and adopts an α-helical conformation in both the inactive and active states of the $M_5$ mAChR (Fig. 4j, Supplementary Fig. 4c). However, upon activation, ICL2 undergoes a significant shift toward the TM core, primarily driven by the inward movement of $F130^{3.52}$, $Y138^{ICL2}$, and $K141^{ICL2}$ whilst $R134^{3.56}$ moves outward (Fig. 4j). These coordinated conformational changes that occur upon receptor activation create the allosteric binding site (Supplementary Fig. 5e,f) enabling VU6007678 binding and explaining why VU6007678 has high affinity for the $M_5$ mAChR active state and low affinity for the $M_5$ mAChR

inactive state. Consequently, this selectivity for the active state explains why VU6007678 acts as a PAM of agonists as opposed to a NAM. Comparison of the allosteric binding site between our active iperoxo-bound structure and the ACh-VU6007678 structure reveals the allosteric binding site is highly similar (Supplementary Fig. 5a,d), indicating the allosteric site is preformed in the $M_5$ mAChR when bound to an orthosteric agonist and that VU6007678 stabilises the $M_5$ mAChR active state.

We performed GaMD simulations on the ACh-VU6007678-bound $M_5$ mAChR-mGα$_q$ complex and with the PAM removed (ACh-bound $M_5$ mAChR-mGα$_q$) to validate the binding pose of VU6007678 and to examine dynamic interactions with the receptor. Both ACh and VU6007678 maintained stable positions in both simulations, with ACh showing minimal fluctuations (RMSD = 1.70 ± 0.28 Å) and VU6007678 displaying moderate mobility while remaining in its binding pocket (RMSD = 4.85 ± 0.56 Å) (Fig. 4h, l, Supplementary Fig. 6).

The VU6007678 binding site contains several residues that vary across mAChR subtypes (Fig. 5a). While $Y68^{2.42}$ is unique to $M_5$, appearing as phenylalanine in other mAChRs, $R134^{3.56}$ and $R146^{4.41}$ are conserved among $M_1$, $M_3$, and $M_5$ but differ in $M_2$ and $M_4$. $V123^{3.45}$ varies between leucine, isoleucine, and valine across $M_1$-$M_4$, while $I149^{4.44}$ alternates between leucine, methionine, and valine in these subtypes. The ICL2 lysine is conserved in $M_1$-$M_3$ but appears as arginine in $M_4$. In contrast, $F126^{3.48}$, $F130^{3.52}$, $T133^{3.55}$, $Y138^{ICL2}$, and $M150^{4.45}$ remain fully conserved across all five mAChR subtypes. This partially non-conserved binding site architecture provided an opportunity to further validate and characterize the binding site through mutagenesis studies and functional assessment using the TruPath G protein activation assay[46].

At WT $M_5$ mAChR, VU6007678 demonstrated robust positive functional cooperativity with ACh signalling and exhibited significant allosteric agonism (Fig. 5b) in a $G_q$ TruPath experiment. Given the

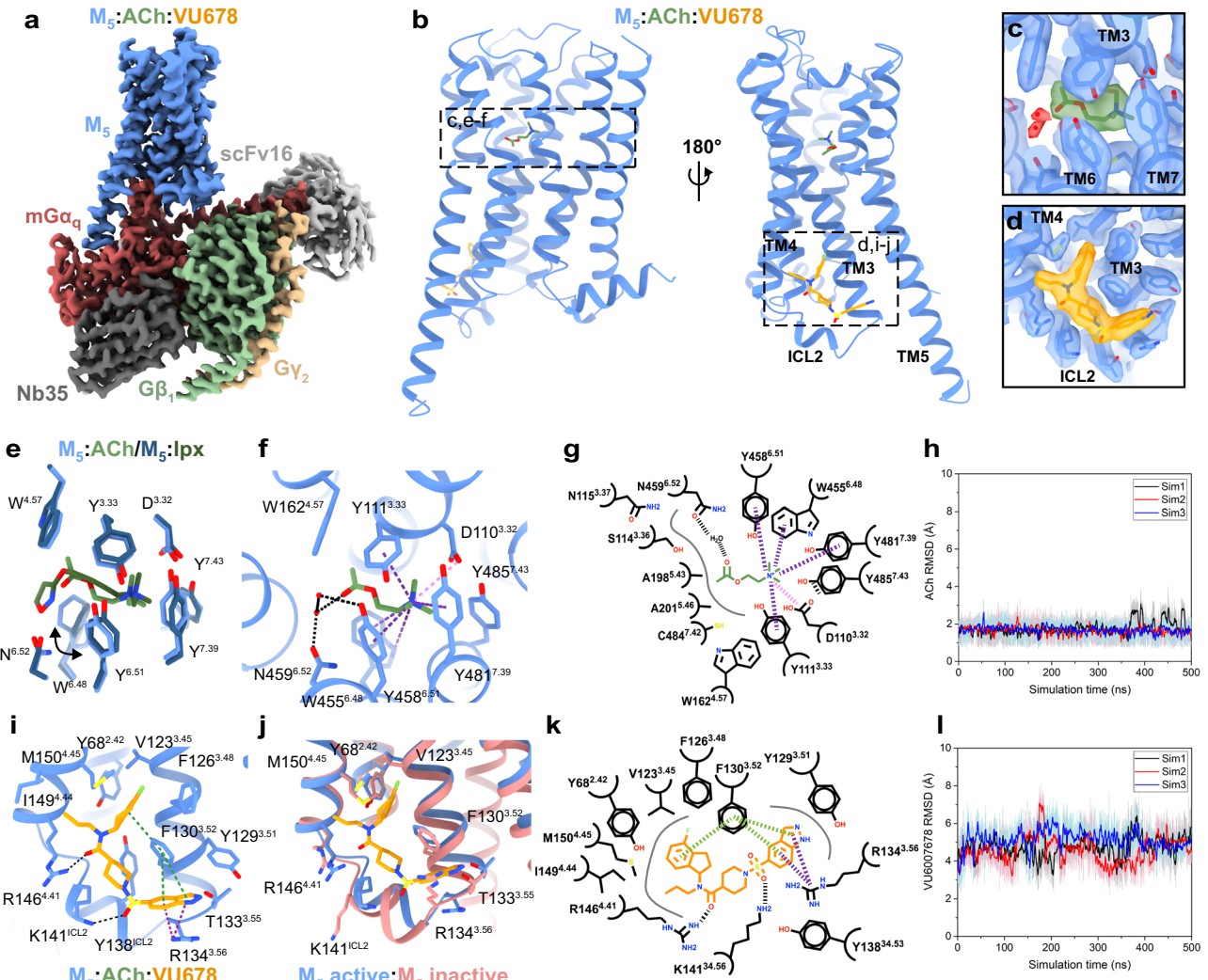

**Fig. 4 | High resolution structure of ACh and VU6007678 bound M₅ mAChR in complex with heterotrimeric G protein. a** Consensus cryo-EM map of M₅ mAChR-mGαₑ/Gβ₁γ₂ complex with ACh and VU6007678 at 2.1 Å resolution (FSC 0.143). Model colouring: light blue (receptor); red (mGαₑ), green (Gβ₁), yellow (Gγ₁), light grey (scFv16), dark grey (Nb35). **b** M₅ mAChR model showing ACh (green) at the orthosteric site and VU6007678 (orange) at the allosteric site between TM3/TM4 and above ICL2. Cryo-EM density (local refined receptor map, contour 0.43) for **c** iperoxo at the orthosteric site and **d** VU6007678 at the allosteric site. **e** Comparison of orthosteric site residues. Stick colouring: light blue (ACh-bound residues), dark blue (Ipx-bound residues), light green (ACh), dark green (Ipx). **f** Interactions of ACh with the orthosteric binding site. Interaction colouring: pink dotted lines (charge-charge), black (hydrogen bonds), purple (cation-π); red spheres (water molecules). **g** 2D interaction plot of ACh with the orthosteric binding site; grey lines (hydrophobic interactions). **h** RMSDs (Å) of ACh relative to the starting conformation during GaMD simulations (three replicates, different colours). **i** VU6007678 allosteric site interactions: black (hydrogen bonds), purple (cation-π), green (π-π). **j** VU6007678 site comparison between the active complex (light blue) and inactive tiotropium-bound M₅ (salmon, PDB:6OL9); VU6007678 (orange). **k** 2D VU6007678 interaction plot. Same colouring as **g** with green dashed lines representing π-π interactions. **l** RMSDs (Å) of VU6007678 relative to the starting conformation during GaMD simulations (three replicates, different colours). Source data are provided as a Source Data file.

selectivity of VU6007678 for the M₅ mAChR over the M₂ mAChR and its extensive ICL2 interactions, we first investigated non-conserved residues at the bottom of TMs 3, 4, and ICL2 by mutating them to their M₂ mAChR counterparts. A construct containing multiple mutations (S131³·⁵³C, I132³·⁵⁴V, R134³·⁵⁶K, R139^ICL2P, A140^ICL2V, P144⁴·³⁹T, R146⁴·⁴¹M, I149⁴·⁴⁴M, G152⁴·⁴⁷A, and L153⁴·⁴⁸A (referred to as the M₅/M₂ swap) caused a complete loss in the ability of VU6007678 to modulate ACh signalling and to display allosteric agonism (Fig. 5c). Individual residue analysis revealed that the R146⁴·⁴¹M mutation reduced both functional modulation and allosteric agonism, while the F130³·⁵²M mutation completely abolished the affinity, functional modulation, and agonism of VU6007678 (Fig. 5d–f, Supplementary Fig. 7). These effects align with our structural data with R146⁴·⁴¹ forming a hydrogen bond with the carboxamide group of VU6007678, and a π-interaction between F130³·⁵² and the indanyl core of VU6007678 (Fig. 4i). Other single

mutations (Y68²·⁴²F, V123³·⁴⁵I, T133³·⁵⁵A, R134³·⁵⁶A, R134³·⁵⁶K and K141^ICL2A) showed no significant changes in affinity, functional modulation, or allosteric agonism (Fig. 5d–f, Supplementary Fig. 7).

Radioligand binding studies of F130³·⁵²M, R146⁴·⁴¹M, and the M₅/M₂ swap showed no significant change in VU6007678 affinity (Fig. 5g, Supplementary Fig. 8, Supplementary Table 4). While the binding cooperativity (log α) was reduced in these constructs (Fig. 5h, Supplementary Fig. 8, Table 4), the values were not significantly different from WT M₅ mAChR, likely due to higher uncertainty in the parameter calculations for the F130³·⁵²M and M₅/M₂ swap constructs. The degree of efficacy modulation (β) by VU6007678 on ACh can be calculated by subtracting the binding modulation (log α) from the functional modulation (log αβ)³⁴. This calculation at WT M₅ mAChR yielded a log β value of 0.4 ± 0.2, indicating that VU6007678's allosteric effect is primarily mediated through binding cooperativity, with a smaller

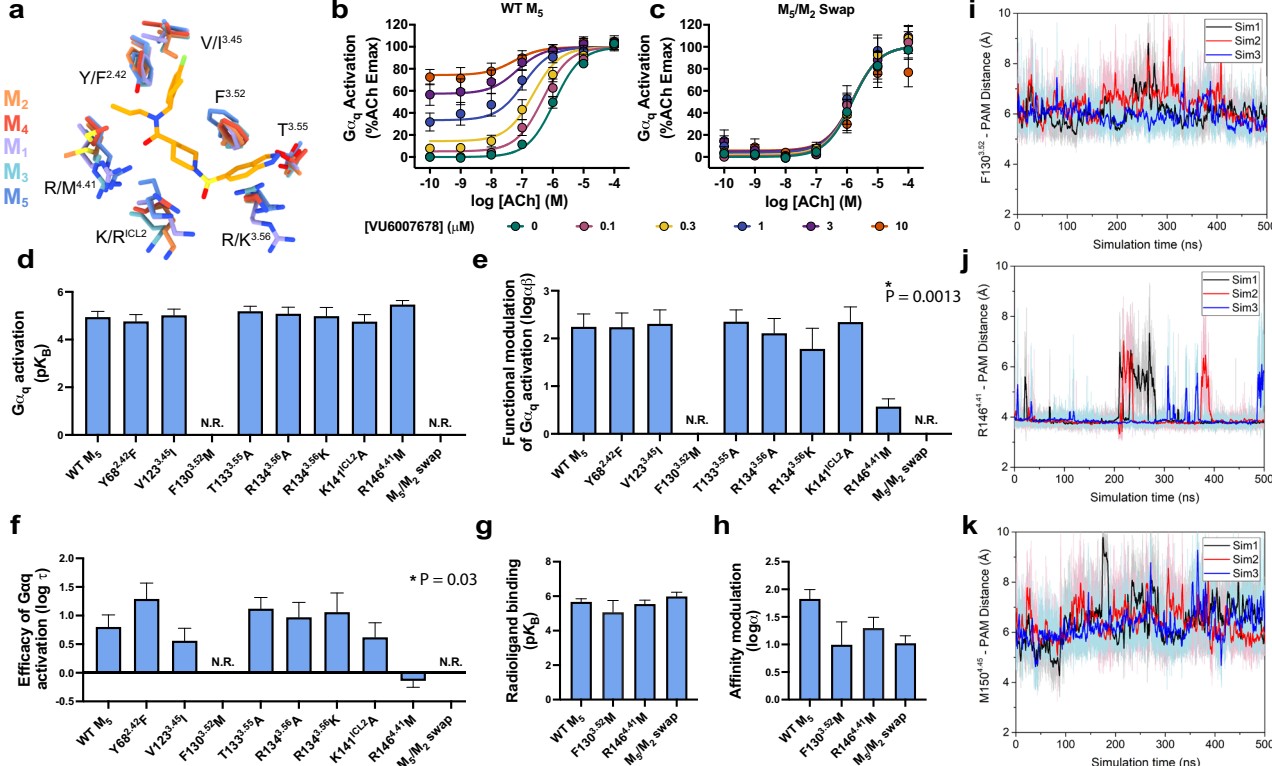

**Fig. 5 | Structural and functional analysis of the VU6007678 allosteric binding site. a** Comparison of VU6007678 allosteric binding site residues (stick representation) across the M$_1$-M$_5$ mAChRs. Colouring: purple (M$_1$; PDB: 6OIJ), orange (M$_2$; PDB: 6OIK), light blue (M$_3$; PDB: 8E9Z), red (M$_4$; PDB: 7TRK), and dark blue (M$_5$; PDB: 9EJZ). VU6007678-ACh interaction in Trupath G$_q$ activation assay at **b** WT M$_5$ mAChR or **c** M$_5$/M$_2$ swap in CHO cells. Data points are mean ± SEM of three to eight individual experiments performed in duplicate. WT M$_5$ mAChR $n = 8$, M$_5$/M$_2$ swap mutant $n = 3$. An operational model of allosterism was fitted to the grouped data to derive the key pharmacological parameters (**d–f**). Pharmacology parameter values were shared across experiments to yield a single best estimate of the mean of each parameter and its associated standard error, as derived from the nonlinear regression algorithm. This global pooled analysis approach ensured model convergence in all instances. *, significantly different

from WT, $p < 0.05$, one-way ANOVA, Dunnett's post hoc test. WT M$_5$ mAChR $n = 8$; M$_5$/M$_2$ swap, F130$^{3.52}$M, T133$^{3.55}$M, R134$^{3.56}$A, R134$^{3.56}$K mutants $n = 3$; Y68$^{2.42}$F, V123$^{3.45}$I, K141$^{ICL2}$A, R146$^{4.41}$M mutants $n = 4$. **g** VU6007678 affinity (pK$_B$) and **h** log affinity cooperativity (logαβ) between ACh and VU6007876 at WT M$_5$ mAChR and mutants from [$^3$H]-NMS equilibrium binding studies. Data points represent mean ± SEM values of pharmacological parameters determined by fitting an allosteric ternary complex model to the grouped data, as described in (**d–f**). All pharmacology parameters (**d–h**) are listed in Supplementary Table 4. WT M$_5$ mAChR $n = 3$, F130$^{3.52}$M, R146$^{4.41}$M $n = 4$, M$_5$/M$_2$ swap $n = 5$. Time courses of distances (Å) from VU6007678 to **i** F130$^{3.52}$, **j** R146$^{4.41}$, **k** M150$^{4.45}$ calculated from GaMD simulations performed with three separate replicates as indicated through different coloured traces. Source data are provided as a Source Data file.

contribution from efficacy modulation. This value could only be compared to the R146$^{4.41}$M mutant, as the F130$^{3.52}$M and M$_5$/M$_2$ swaps showed no functional modulation. The R146$^{4.41}$M mutant yielded a log β value of −0.7 ± 0.3, suggesting impaired efficacy modulation by VU6007678. These findings are supported by our previous characterisation of VU6007678 in receptor alkylation studies, which revealed modest efficacy modulation in functional IP one assays[29]. Interestingly, this observation was unique to VU6007678 in the structure-activity relationship (SAR) study. Collectively, these data highlight the importance of the VU6007678 binding site residues in mediating both functional and efficacy modulation. VU6007678 directly interacts with F126$^{3.48}$, Y129$^{3.51}$, F130$^{3.52}$, residues adjacent to D127$^{3.49}$ and R128$^{3.50}$ that form the DRY activation motif. Additionally, VU6007678 mediated stabilisation of ICL2 facilitates interactions between L136$^{ICL2}$ and R139$^{ICL2}$ with the α5 and αN helices of the G protein, respectively. Together, these interactions explain how VU6007678 functions as an Ago-PAM by stabilising key receptor activation motifs and G protein interactions.

Further validation of the allosteric binding site through GaMD simulations revealed key interactions between VU6007678 and receptor residues. The PAM formed stable interactions with F130$^{3.52}$, R146$^{4.41}$, and M150$^{4.45}$ (Fig. 5i–k), consistent with both the cryo-EM

structure and mutagenesis data. In contrast, hydrogen bond interactions between VU6007678 and receptor residues R134$^{3.56}$, T133$^{3.55}$, and K141$^{ICL2}$ showed greater variability with larger distances and higher fluctuations (Supplementary Fig. 9), aligning with our experimental observations where mutations of these residues did not significantly affect VU6007678 function. In addition, comparison of the ACh- and ACh-VU6007678-bound GaMD simulations revealed that VU6007678 led to an increase in the helical turn of ICL2 by one residue, and reduced fluctuations in ICL2 residues K141$^{ICL2}$ and R142$^{ICL2}$ (Supplementary Fig. 6g, h). Distinct conformations of ICL2 have been linked to G protein bias[47] and allosteric activation[48]. Together, these results highlight how VU6007678 engages key residues involved in the activation of the M$_5$ mAChR, specifically via stable interactions with F130$^{3.52}$ and R146$^{4.41}$, while stabilizing part of ICL2 via interactions with K141$^{ICL2}$ and R142$^{ICL2}$.

## Discussion

Allosteric modulators for the mAChRs have long been pursued for selective targeting of a specific mAChR subtype. Whilst selective allosteric modulators for the M$_5$ mAChR have been discovered, the development and application of these have lagged behind those of allosteric modulators selective for other mAChRs, particularly the M$_1$

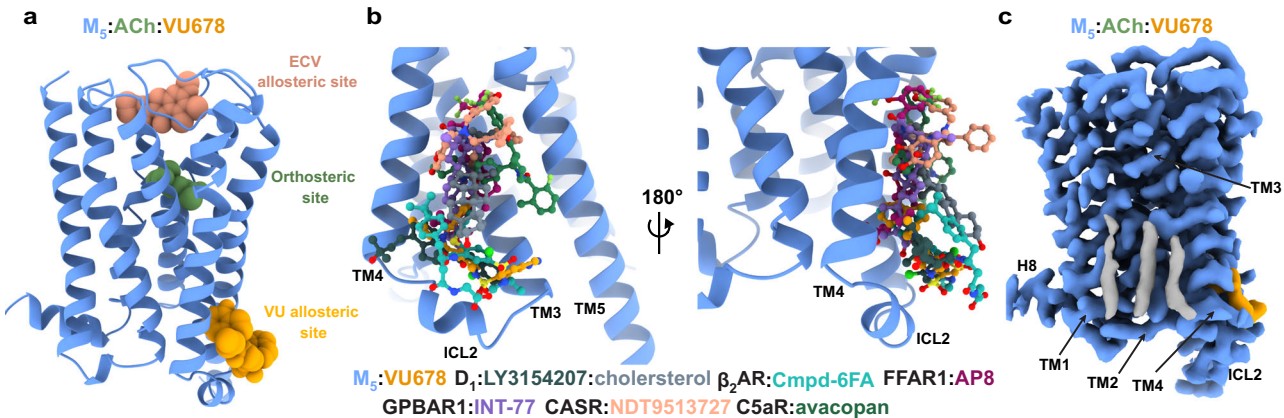

**Fig. 6 | Allosteric sites at the mAChRs. a** Model of the ACh-VU6007678 bound M5 mAChR showing the allosteric binding site discovered for VU6007678 (orange spheres) and the ECV allosteric site occupied by LY2119620 (salmon spheres). ACh is shown as green spheres in the orthosteric binding site. **b** Allosteric modulator structures overlaid on the VU allosteric site, with ligands shown as ball-and-stick models. D1:LY3154207:cholersterol in dark slate grey and slate grey, respectively (PDB:7X2F). β2AR:Cmpd-6FA in light sea green (PDB:6N48). FFAR1:AP8 in violet (PDB:5TZY). GPBAR1:INT-177 in purple (PDB:7CFN). CASR:NDT9513727 in salmon (PDB:6C1Q). C5aR:avacopan in green (PDB:6C1R). **c** Cryo-EM density (contour level 0.47) for lipid molecules observed at the bottom of TM2,3,4.

and M4 mAChRs. This partly reflects the limited structural knowledge of the M5 mAChR overall, as well as the specific lack of insight into the allosteric binding site for M5-selective PAMs. Here we present a cryo-EM structure of the M5 mAChR bound to the orthosteric agonist iperoxo that 'completes' the active state structure ensemble for all five mAChR subtypes. Initial attempts to solve the co-bound structure of an allosteric modulator with ML380 and iperoxo were unsuccessful. Yet through the use of analytical pharmacology to determine the optimum orthosteric and allosteric ligand combination and improved biochemical techniques, we obtained a high-resolution structure of the M5 mAChR co-bound to the endogenous orthosteric agonist ACh and the selective PAM VU6007678. Our data begins to explain several facets of selective allosteric modulation at the M5 mAChR, including 1) the historic difficulties in developing subtype-selective PAMs for this mAChR subtype, 2) how changes to ML380 led to the improved PAM VU6007678, 3) subtype selectivity, and 4) the mechanism of action.

The in vivo translation of M5 mAChR-selective PAMs has been hindered by issues related to DMPK and suboptimal partition coefficients (Kp). The observation that the allosteric binding site for VU6007678 is located in the transmembrane (TM) bundle partially explains this, as modulators must display a high degree of lipophilicity to reach this allosteric binding site. Despite this, the VU6007678 scaffold offers numerous opportunities for modification, and our structure will aid in the process of developing improved allosteric modulators, as it provides information on the molecular interactions that occur between VU6007678 and the M5 mAChR at its allosteric binding site. Specifically, Y68[2.42] is the only residue within the allosteric binding site that differs across all M1-M4 mAChR subtypes. It may therefore be possible to introduce and optimize the presence of various polar functional groups on the indanyl core to promote hydrogen bonding between the ligand and receptor and enhance affinity whilst reducing the compound's lipophilicity. Note that the compounds reported in the previous SAR series all had a fluorine functional group attached to the indanyl core[26]. Enhancing the selectivity of PAMs for the M5 mAChR will be crucial, in part, to the future clinical success of M5 mAChR PAMs. All M5 PAMs discovered to date display activity at the M1 and M3 mAChRs, whilst they are most selective against the M2 and M4 mAChRs, where they display very little to no activity. Analysis of the residues within the VU6007678 allosteric binding site may explain the basis for this, as R134[3.56] and R146[4.41] are fully conserved between the M5 mAChR and M1/M3 mAChRs and non-conserved between the M5 mAChR and the M2/M4 mAChRs.

Despite the absence of an ML380-bound M5 mAChR structure, given that VU6007678 is a derivative of ML380, predictions on how ML380 interacts with this site can be made. This allows for an explanation of how the changes to VU6007678 led to an improved PAM. The extension of the ethyl present in ML380 into a propyl group at VU6007678 gives rise to an extra interaction with I149[4.44]. In addition, the substitution of the trifluoromethylbenzyl group at ML380 with the indanyl group gives rise to more hydrophobic interactions and greater engagement with the top of the allosteric site consisting of V123[3.45], F126[3.48] and M150[4.45] where edge-to-face π-interaction takes place with F126[3.48].

Our structures offer key insights into the mechanism of action for PAMs at the M5 mAChR. The allosteric binding site of VU6007678 is distant from the orthosteric site but positioned near the highly conserved DRY activation motif, suggesting that PAM binding stabilizes the active state of the M5 mAChR. This is in contrast to other allosteric modulators at the mAChRs. The mAChR family has served as a model receptor family for the study of allosteric modulation at GPCRs[19]. Despite a wealth of mAChR allosteric modulators with different scaffolds available, this is only the second site to be confirmed by structural biology studies at the mAChRs[33,34] (Fig. 6a). All PAMs discovered to date bind to the mAChRs at what is termed the ECV allosteric site in a solvent accessible vestibule on top of the orthosteric binding site. Here, PAMs exert their mechanism of action through two mechanisms: (1) trapping the orthosteric ligand in the orthosteric site through stabilising the active state and (2) sterically hindering orthosteric ligand dissociation[49,50]. Due to the distant location of the VU6007678 binding site to the mAChR orthosteric site, our findings suggest that PAMs at this site act exclusively by stabilising the active state. Numerous allosteric modulators at other GPCRs have been structurally confirmed to bind at the same site as VU6007678 at the M5 mAChR, with diverse mechanisms of action (Fig. 6b)[51–59]. Structures of the M5 mAChR have an ordered ICL2 in an α-helical conformation in both the inactive and active states, irrespective of VU6007678 binding, although GaMD simulations suggest VU6007678 binding stabilizes an additional turn of the ICL2 α-helix. Conversely, at the β2-adrenergic receptor (β2AR), ICL2 is disordered in the inactive state and becomes α-helical upon receptor activation. Similar to VU6007678, the β2AR PAM Cmpd-6FA stabilises ICL2, but unlike VU6007678, Cmpd-6FA does influence the allosteric site by changing the orientation of interacting residues[52]. In contrast, at the free fatty acid receptor 1 (FFAR1), ICL2 only becomes ordered in the presence of the PAM AP8[48,56]. Together, these structures

illustrate that despite all binding to the extrahelical interface of TM2,3,4, the manner through which they engage with ICL2 differs.

Hydrogen-deuterium exchange mass spectrometry and NMR studies suggest that the conformational dynamics of ICL2 are important for receptor interactions with the Gα protein and G protein specificity[47,60–63]. Thus, targeting the ICL2 allosteric site could facilitate the design of biased allosteric modulators[64]. Interestingly, we observed that the K141$^{ICL2}$A mutant increased the $E_{max}$ of ACh-mediated $G_q$ activation in the presence of VU6007678 compared to WT M$_5$ mAChR. In GaMD simulations, K141$^{ICL2}$ was relatively dynamic, though less so when bound to VU6007678. Mutation of K141$^{ICL2}$ to alanine likely alters the conformational dynamics of ICL2, leading to increased signalling when bound to VU6007678. Future studies investigating the dynamics of ICL2 at different GPCRs when bound to PAMs will aid our understanding of G protein signalling and the design of biased allosteric modulators.

Though less reported, NAMs at the extrahelical interface of TMs 3,4 are reported to inhibit receptor activation by inhibiting helical movement required for receptor activation[58,59]. NAM structures identified to date suggest NAMs occupy a more confined site at the centre of the TM2,3,4 interface. This raises an interesting possibility: NAMs may primarily target allosteric sites at the centre of the extrahelical interface to restrict the conformational changes necessary for receptor activation, whereas PAMs engage closer to the cytosol, ICL2, and activation motifs to stabilize the active state. The discovery of a methionine (M$^{4.45}$) residue within the VU6007678 allosteric binding site enables deeper investigation into allosteric mechanisms. Spectroscopic analysis of PAM activity at a methionine-labelled M$_5$ mAChR, as done for LY2119620 at the M$_2$ mAChR[65], will further elucidate their mode of action.

At the bottom of the TM2,3,4 interface of the ACh-VU6007678 structure, we observed well-defined densities that likely correspond to three cholesterol molecules (Fig. 6c). Note that in our iperoxo structure, this site was occupied by a partial density that we cautiously hypothesised could be ML380. Considering the much higher resolution of the ACh-VU6007678 structure, and assignment of the VU6007678 binding site at the bottom of the extrahelical interface at TMs3-4, it is likely that the density present in the iperoxo-bound structure represents cholesterol molecules. Especially given that this site is recognised as a common cholesterol site within class A GPCRs[44,66] and that neurosteroids and steroid hormones, including derivatives of cholesterol, have displayed modulatory properties at the M$_5$ mAChR[67,68]. Due to the lower density and map quality in this region, it was difficult to conclusively assign this density to either ML380 or cholesterol in the iperoxo structure, highlighting the map quality required to assign density to small molecules. Particularly at extrahelical regions where a number of lipid and cholesterol molecules may be present.

Altogether and more broadly, the discovery of this extrahelical allosteric binding site at the M$_5$ mAChR adds to the knowledge of allostery at the mAChR, specifically on how allosteric modulators engage with the mAChRs, and provides an additional avenue through which to target these highly conserved proteins.

## Methods

### Chemical Probe Statement

ML380 is a positive allosteric modulator for the $G\alpha_{q/11}$ coupled mAChRs subtypes (M$_1$, M$_3$, M$_5$) with a binding affinity for the ACh-bound M$_5$ mAChR of ~575 nM (determined in this study and others[25,28]). VU6007678 is an improved chemical analogue with ~30 nM affinity (determined in this study and others[29]) for the ACh-bound M$_5$ mAChR and 150 nM affinity for the M$_1$ and M$_3$ mAChRs (5-fold selectivity, determined in other studies[29]). We used ML380 and VU6007678 at concentration ranges that are appropriate for in vitro pharmacology studies and cryo-EM. In the development of ML380, the (S) and (R)

enantiomers were tested for activity, with the (S)-enantiomer being inactive[25]. Both ML380 and VU6007678 are the (R)-enantiomers. Both probes are active on receptors expressed in cells. We did not use the inactive (S)-enantiomer probe in this study, as our study was designed for the validation of the allosteric binding site using cryo-EM. As previously reported, these molecules likely need optimisation before being used in animal models[25,28,29].

### Receptor & G protein co-expression

A modified M$_5$ mAChR construct was used where residues 237-421 of ICL3 were removed and HA signal sequence and anti-Flag epitope tag were added to the N terminus. Modified M$_5$ mAChR was fused to a mGα$_q$ chimeric construct that is a mini-Gα$_s$ substituted with Gα$_q$ residues at the receptor interface and the αN of Gα$_i$[39,40]. A 3 C protease recognition site and GGGS linker were used to separate the receptor and G protein. The fused M$_{5ΔICL3}$mGα$_q$/construct was cloned into a pFastbac baculovirus transfer vector. G protein β$_1$ and γ$_2$ subunits were cloned into a pVL1392 baculovirus transfer vector with the β subunit modified to contain a carboxy (C)-terminal 8× histidine tag. *Trichoplusia ni* (Hi5) insect cells were grown in ESF 921 serum-free media (Expression Systems) and infected at a density of $4.0 \times 10^6$ cells per millilitre with a 1:1 ratio of M$_{5ΔICL3}$mGα$_q$ to Gβ$_1$γ$_2$ viruses and shaken at 27 °C for 48–60 hours. Cells were harvested by centrifugation, and the cell pellets were flash-frozen using liquid nitrogen and stored at −80 °C. Sf9 and Hi5 cells were not tested for mycoplasma.

### Single-chain stabilising fragment expression and purification

A single-chain construct of Fab16 (scFv16)[41], tagged with an 8× histidine sequence at the C-terminus, was cloned into a modified pVL1392 baculovirus transfer vector for secreted expression in *Trichoplusia ni* (Hi5) insect cells (Expression Systems). The cells were cultured in serum-free ESF 921 media (Expression Systems), infected at a density of $4.0 \times 10^6$ cells per millilitre, and incubated with shaking at 27 °C for 48-72 hours. For purification, the pH of the supernatant from the baculovirus-infected cells was adjusted with Tris pH 8.0. Chelators were neutralized by adding 5 mM CaCl$_2$ and stirring the solution for 1 hour at 25 °C. Precipitates were cleared by centrifugation, and the supernatant was then applied to Ni-NTA resin. The column was washed with a solution of 20 mM Hepes pH 7.5, 50 mM NaCl, and 10 mM imidazole, followed by a second wash containing the same buffer with 100 mM NaCl. The scFv16 protein was eluted using the low-salt buffer with 250 mM imidazole. SDS-PAGE with Coomassie staining was used to assess the purity of the eluted protein. Finally, the sample was concentrated, flash-frozen in liquid nitrogen, and stored at −80 °C.

### Nb35 expression and purification

Nb35 was expressed in the periplasm of the BL21(DE3) Rosetta *Escherichia coli* cell line using an autoinduction approach[69]. Transformed cells were cultured at 37 °C in a modified ZY medium containing 50 mM phosphate buffer (pH 7.2), 2% tryptone, 0.5% yeast extract, 0.5% NaCl, 0.6% glycerol, 0.05% glucose, and 0.2% lactose, with 100 µg/ml carbenicillin and 35 µg/ml chloramphenicol. When the culture reached an OD600 of 0.7, the temperature was reduced to 20 °C for approximately 16 hours, after which cells were harvested by centrifugation and stored at −80 °C. To purify, cells were lysed in ice-cold buffer (0.2 M Tris pH 8.0, 0.5 M sucrose, 0.5 M EDTA) at a ratio of 1 g of cell pellet to 5 mL of lysis buffer for 1 hour at 4 °C. 2x volume of ice-cold MQ was added and incubated for an additional 45 minutes at 4 °C. Lysate was centrifuged to remove cell debris, and supernatant containing Nb35 was spiked with 20 mM HEPES pH 7.5, 150 mM NaCl, 5 mM MgCl$_2$, 5 mM Imidazole and applied to Ni-NTA resin, followed by 90 minutes incubation at 4 °C. The column was washed with 40 column volumes (CV) of wash buffer (20 mM HEPES pH 7.5, 500 mM NaCl, 5 mM Imidazole) and eluted with elution buffer (20 mM HEPES pH 7.5, 100 mM NaCl, 250 mM Imidazole). Elute was concentrated and stored at −80 °C.

## Complex purification

**Iperoxo-bound M$_5$ mAChR-mGα$_q$ complex.** M$_{5\Delta ICL3}$-mGα$_q$ co-expressed with Gβ$_1$γ$_2$ was thawed and lysed in 20 mM HEPES pH 7.4, 5 mM MgCl$_2$, 1 µM Ipx and protease inhibitors (500 µM PMSF, 1 mM LT, 1 mM benzamidine). The sample was rotated at room temperature for 15 minutes and spiked with apyrase towards the end of the 15 minutes. The pellet was spun down by centrifugation and solubilized in 20 mM HEPES pH 7.4, 100 mM NaCl, 5 mM MgCl$_2$, 5 mM CaCl$_2$, 0.5% LMNG (Anatrace, Maumee, OH, USA), 10 µM Ipx, and protease inhibitors (500 µM PMSF, 1 mM LT, 1 mM benzamidine). The resuspended pellet was homogenised in a Dounce homogeniser, and complex formation was initiated through addition of scFv16. The sample was incubated with stirring at 4 °C for 2 hours followed by centrifugation to remove insoluble material. Solubilised complex was bound to equilibrated M1 anti-Flag affinity resin through batch binding at room temperature for 1 hour. The resin was packed into a glass column and washed with 20 mM HEPES pH 7.4, 100 mM NaCl, 5 mM MgCl$_2$, 5 mM CaCl$_2$, 0.01% LMNG, and 1 µM Ipx until no more protein was coming off the column as determined by Bradford. Complex was eluted using 20 mM HEPES pH 7.4, 100 mM NaCl, 5 mM MgCl$_2$, 0.01% LMNG, and 1 µM Ipx in the presence of 5 mM EDTA and 0.1 mg/ml Flag peptide. Eluted complex was concentrated in an Amicon Ultra-15 100 kDa molecular mass cut-off centrifugal filter unit (Millipore, Burlington, MA, USA) and purified by size exclusion chromatography (SEC) on a Superdex 200 Increase 10/300 GL (Cytiva, Marlborough, MA, USA) in 20 mM HEPES pH 7.4, 100 mM NaCl, 5 mM MgCl$_2$, 0.01% LMNG, and 1 µM Ipx. Fractions containing complex (as determined by SDS-page and Coomassie staining) were pooled and concentrated to 3.5 mg/mL, flash frozen using liquid nitrogen and stored at −80 °C.

**ACh-VU6007678-bound M$_5$ mAChR-mGα$_q$ complex.** The ACh-VU6007678 M$_5$ mAChR sample was purified in an identical manner with the following changes; (1) complex formation was initiated through addition of scFv16 and Nb35, (2) ACh was included in all buffers at 100 µM until SEC where a concentration of 10 µM was utilised, (3) VU6007678 was present throughout the purification at a concentration of 10 µM. The final purified product was concentrated to 18 mg/mL.

## Vitrified sample preparation and data collection

For the iperoxo-M$_5$ mAChR sample, EMAsian - TiNi 200 mesh 1.2/1.3 grids were glow discharged using Pelco EasyGlow for 90 seconds with 15-mA current. Prior to grid freezing, the iperoxo-M$_5$ mAChR sample was spiked with 30 µM of ML380 and incubated overnight at 4 °C. 3 µL of this spiked sample was applied and flash frozen in liquid ethane using a Vitrobot markIV with a blot force of 4 and blot time of 2 seconds at 100% humidity and 4 °C. Data were collected on a Titan Krios (Thermo Fisher Scientific) 300 kV electron microscope equipped with a K3 detector, 50 µm C2 aperture, no objective aperture inserted, indicated magnification x 130 000 in nanoprobe TEM mode, a slit width of 10 eV, pixel size 0.65 Å, exposure rate 10.57 counts per pixel per second, exposure time 2.68 s, total exposure 60 e Å$^{-2}$, and 60 frames. In total, 9104 movies were collected.

For the ACh-VU6007678 M$_5$ mAChR sample, 3 µL of sample was applied to a glow-discharged (15 mA, 180 s) UltrAufoil R1.2/1.3 300 mesh holey grid (Quantifoil) and was frozen in liquid ethane using a Vitrobot mark IV (Thermo Fisher Scientific) at 100% humidity and 4 °C with a blot time of 2 s and blot force of 10. The sample was collected similarly, except at 105kX magnification with a pixel size of 0.82 Å, and 7489 movies were collected.

## Image Processing

For the iperoxo-M$_5$ mAChR sample, 9104 movies were collected and adjusted for beam-induced motion by MotionCor2[70]. Non-dose-weighted micrographs were used for CTF estimation using Gctf[71].

8341 micrographs were identified as having a ctf fit resolution below 4 Å, and these were selected for further examination. 3,833,583 particles were autopicked using Gautomatch (https://www2.mrc-lmb.cam.ac.uk/research/locally-developed-software/zhang-software/#gauto). The particles were extracted with relion-3.1[72] and then imported into CryoSparc[73] for rounds of 2D classification, ab initio 3D and 3D refinement to obtain a 3.04 Å model. Particles were taken to Relion3.1 for polishing and subsequent 3D refinement back in Cryosparc yielded a final model of 2.75 Å based on the gold standard Fourier shell correlation cut-off of 0.143 from 426,714 particles. A further local refinement was performed to generate a receptor-focused map (2.67 Å).

The ACh-VU6007678 M$_5$ mAChR sample was processed similarly. 7489 movies were collected and adjusted for beam-induced motion by MotionCor2[70]. Non-dose weighted micrographs were used for CTF estimation using Gctf[71]. 4,436,835 particles were autopicked using Gautomatch (https://www2.mrc-lmb.cam.ac.uk/research/locally-developed-software/zhang-software/#gauto). The particles were extracted with relion-3.1[72] and then imported into CryoSparc[73] for rounds of 2D-classification and heterogeneous refinement. A set of particles (~900k) was polished in relion3.1 and a final round of 3D-classification (no alignment) was performed. This final set of 418,794 particles was finally subjected to non-uniform refinement with CTF-refinement in CryoSparc, resulting in a final map of 2.06 Å based on the gold standard Fourier shell correlation cut-off of 0.143. A further local refinement was performed to generate a receptor-focused map (2.11 Å).

## Model building and refinement

An initial receptor model was generated from the cryo-EM structure of the M$_4$ mAChR receptor (PDB: 7TRP). An initial model for the G protein (mGα$_q$:Gβ$_1$Gγ$_2$:ScFv16) was generated from the CCK1:mGα$_q$ complex (PDB: 7MBY). Initial models were placed in the EM maps using UCSF ChimeraX[74] and rigid-body-fit using PHENIX[75]. Models were refined with iterative rounds of manual model building in Coot[76] and ISOLDE, and real-space refinement in PHENIX. Ligands ACh and iperoxo were obtained from the monomer library, while the initial model and restraints for VU6007678 were generated using the GRADE web server (https://grade.globalphasing.org). Model validation was performed with MolProbity[77] and the wwPDB validation server[78]. Figures were generated with UCSF ChimeraX and PyMOL (Schrödinger).

## Cell culture

FlpIn Chinese hamster ovary (CHO) cells (Thermo Fisher Scientific) stably expressing M$_5$ mAChR constructs were cultured at 37 °C in 5% CO$_2$ using Dulbecco's modified Eagle's medium (DMEM; Invitrogen) supplemented with 5% foetal bovine serum (FBS; ThermoTrace). At confluence, media was removed, and cells were washed with phosphate-buffered saline (PBS) and harvested from tissue culture flasks using Versene (PBS with 0.02% EDTA). The cells were pelleted by centrifugation at 350 g for three minutes and then resuspended in DMEM with 5% FBS. Subsequently, the cells were either plated for an assay or reseeded into a tissue culture flask. CHO cells were regularly tested to ensure they were free from mycoplasma.

## Inositol Monophosphate (IP1) Accumulation Assay

FlpIn CHO cells stably expressing either WT or mutant hM$_5$ mAChR were seeded in clear, flat bottom 96-well plates at a density of 10,000–25,000 cells per well (depending on the cell line) one day prior to the assay. The optimal cell density for each line was chosen based on achieving an IP1 response that fell within the linear range of the IP1 standard curve. On the assay day, the medium was replaced with stimulation buffer (Hanks' balanced salt solution (HBSS) containing 10 mM HEPES, 1.3 mM CaCl$_2$, and 30 mM LiCl, pH 7.4) and allowed to incubate for 60 minutes at 37 °C before ligand stimulation. After this pre-incubation, the buffer was replaced, and cells were exposed to

ligands for 60 minutes at 37 °C in a 5% $CO_2$ atmosphere, with a total assay volume of 100 μL. Following the 60-minute stimulation, ligands were removed by rapid removal of buffer. Cells were lysed by freeze-thawing in 30 μL of stimulation buffer. IP1 accumulation was then quantified using the HTRF IP-One assay kit (Cisbio), with fluorescence measured on an EnVision multilabel plate reader (PerkinElmer).

## TruPath – G protein Activation Assay

Upon reaching 60-80% confluence, FlpIn CHO cells stably expressing WT or mutant $hM_5$ mAChR were transiently transfected using Poly-ethylenimine (PEI; Sigma-Aldrich). For each well, 10 ng of each plasmid (pcDNA5/FRT/TO-$G\alpha_q$-RLuc8, pcDNA3.1-$\beta_3$, and pcDNA3.1-$G\gamma_9$-GFP2) was added in a 1:1:1 ratio, totalling 30 ng. These plasmids were gener-ously provided by Prof. Bryan Roth from the University of North Car-olina. The cells were then plated at 30,000 cells per well into 96-well Greiner CELLSTAR white-walled plates (Sigma-Aldrich). After 48 hours, the cells were washed with 200 μL PBS and replaced with 1x HBSS supplemented with 10 mM HEPES. The cells were incubated for 30 minutes at 37 °C before adding 10 μL of 1.3 μM Prolume Purple coelenterazine (Nanolight Technology, Pinetop, AZ). Following a fur-ther 10-minute incubation at 37 °C, bioluminescence resonance energy transfer (BRET) measurements were performed using a PHERAstar FSX plate reader (BMG Labtech) with 410/80-nm and 515/30-nm filters. Four baseline measurements were taken before adding drugs or vehicle, bringing the final assay volume to 100 μL, followed 10 more minutes of readings. The BRET signal was calculated as the ratio of 515/30-nm emission to 410/80-nm emission. This ratio was vehicle-corrected using the initial four baseline measurements and then baseline-corrected again using the vehicle-treated wells. Data were normalized to the maximum ACh response to allow for grouping of results.

## Radioligand Binding

FlpIn CHO cells stably expressing WT $hM_5$ mAChR or mutants were plated at 25,000 cells per well in 96-well isoplates (PerkinElmer Life Sciences) and incubated overnight at 37 °C in a 5% $CO_2$ incubator. The following day, the cells were washed with PBS and incubated in 20 mM HEPES, 100 mM NaCl, 10 mM $MgCl_2$, pH 7.4. For saturation binding experiments, the cells were incubated with varying concentrations of the orthosteric antagonist [$^3$H]-N-methylscopolamine ([$^3$H]-NMS; spe-cific activity, 70 Ci/mmol, Perkin Elmer) in a final volume of 100 μL for 6 hours at room temperature. For interaction experiments between orthosteric agonist and allosteric modulator, competition binding was performed between a $K_D$ concentration of [$^3$H]-NMS and varying con-centrations of an orthosteric drug in the presence of different con-centrations of an allosteric modulator, in a total volume of 100 μL binding buffer. For all experiments, non-specific binding was defined using 10 μM of atropine. The assay was terminated by the rapid removal of the radioligand, followed by two 100 μL washes with ice-cold 0.9% NaCl buffer. Radioactivity was measured by adding 100 μL of Optiphase Supermix scintillation fluid (PerkinElmer) and counted using a MicroBeta$^2$ Plate Counter (PerkinElmer Life Sciences).

## Data analysis

All data were analysed using GraphPad Prism 10 (GraphPad Software, San Diego, CA). The interaction between orthosteric agonist and allosteric modulator in functional assays was analysed using an operational model of allosterism to determine functional modulation (log αβ) and affinity (p$K_B$) parameters[36]. Radioligand saturation bind-ing experiments with [$^3$H]-NMS to determine $B_{max}$ and p$K_D$ values were determined using a one site – specific binding equation in Prism 10[31]. For the radioligand binding interaction of orthosteric agonist with various concentrations of allosteric modulator, the data were fit to an allosteric ternary complex model to derive p$K_B$ and α binding coop-erativity parameters[79]. All affinity, potency, cooperativity, and efficacy

parameters were estimated as logarithms. To ensure model con-vergence in all instances, allosteric parameters were determined from the respective allosteric models, which were globally fitted to all individual datasets for each allosteric modulator at each receptor construct. These parameters were constrained to be shared across these datasets. The resulting reported parameter estimates thus represent the least-squares best-fit value of each parameter, with its associated standard error as reported from the nonlinear regression programme, based on the pooled analysis of the individual datasets. Statistical analysis between different treatment conditions was per-formed using one-way ANOVA, with a p-value of <0.05 considered significant.

## Gaussian accelerated Molecular Dynamics (GaMD) simulations anD Simulation Analysis

GaMD is an enhanced sampling method that works by adding a har-monic boost potential to reduce the system energy barriers. Details of the method have been described in previous studies[80,81]. A summary is provided here. Consider a system with $N$ atoms at positions $\vec{r} = \{\vec{r}_1, \cdots, \vec{r}_N\}$. When the system potential $V(\vec{r})$ is lower than a reference energy $E$, the modified potential $V^*(\vec{r})$ of the system is calculated as:

$$V^*(\vec{r}) = V(\vec{r}) + \Delta V(\vec{r}), \tag{1}$$

$$\Delta V(\vec{r}) = \begin{cases} \frac{1}{2} k (E - V(\vec{r}))^2, & V(\vec{r}) < E \\ 0, & V(\vec{r}) \geq E \end{cases} \tag{2}$$

where $k$ is the harmonic force constant. The two adjustable parameters $E$ and $k$ are automatically determined based on three enhanced sam-pling principles. The reference energy needs to be set in the following range:

$$V_{max} \leq E \leq V_{min} + \frac{1}{k}, \tag{3}$$

where $V_{max}$ and $V_{min}$ are the system minimum and maximum potential energies. To ensure that Eq. (3) is valid, $k$ has to satisfy: $k \leq \frac{1}{V_{max} - V_{min}}$ Let us define $k \equiv k_0 \frac{1}{V_{max} - V_{min}}$, then $0 < k_0 \leq 1$. The standard deviation of $\Delta V$ needs to be small enough (i.e., narrow distribution) to ensure proper energetic reweighting: $\sigma_{\Delta V} = k \left( E - V_{avg} \right) \sigma_V \leq \sigma_0$ where $V_{avg}$ and $\sigma_V$ are the average and standard deviation of the system potential ener-gies, $\sigma_{\Delta V}$ is the standard deviation of $\Delta V$ with $\sigma_0$ as a user-specified upper limit (e.g., $10k_BT$) for proper reweighting. When $E$ is set to the lower bound $E=V_{max}$, $k_0$ can be calculated as:

$$k_0 = \min(1.0, k_0') = \min \left( 1.0, \frac{\sigma_0}{\sigma_V} \frac{V_{max} - V_{min}}{V_{max} - V_{avg}} \right), \tag{4}$$

Alternatively, when the threshold energy $E$ is set to its upper bound $E = V_{min} + \frac{1}{k}$, $k_0$ is set to:

$$k_0 = k''_0 \equiv \left( 1 - \frac{\sigma_0}{\sigma_V} \right) \frac{V_{max} - V_{min}}{V_{max} - V_{avg}}, \tag{5}$$

if $k''_0$ is found to be between $0$ and $1$. Otherwise, $k_0$ is calculated using Eq. (4).

The cryo-EM structures of the ACh-VU6007678-bound $M_5$ mAChR-$mG\alpha_q$ complex and the iperoxo-bound $M_5$ mAChR-$mG\alpha_q$ complex were used to set up initial simulation systems. For simulations with ACh-$M_5$ mAChR-$mG\alpha_q$, VU6007678 was manually removed from the structure. The initial model of the iperoxo-ML380-bound $M_5$ mAChR-$mG\alpha_q$ complex was created by docking ML380 into the bot-tom of the TM2,3,4 interface. As in our previous studies[34], the

intracellular loop 3 (ICL3) of the receptor and the α-helical domain of the G protein, which were missing in the cryo-EM structure, were not modelled. The MD simulation systems were prepared by inserting the ACh-VU6007678-bound $M_5$ mAChR-mGα$_q$ and the iperoxo-bound $M_5$ mAChR-mGα$_q$ complexes into a POPC (palmitoyl-2-oleoyl-sn-glycero-3-phosphocholine) lipid bilayer using VMD (Visual Molecular Dynamics). In each simulation system, the protein and lipid bilayer were solvated with TIP3P water molecules in a box of 12.5 nm x 12.5 nm x 14.0 nm with the periodic boundary condition. The system charge was neutralized with 150 mM NaCl. The AMBER FF14SB force field[82] was applied for the proteins, while the AMBER LIPID21 force field[83] was used for the lipids. The general amber force field (GAFF2) parameters[84] for ACh, iperoxo, ML380, and VU6007678 were generated with ANTECHAMBER. The two simulation systems were first energy minimized for 5,000 steps with constraints on the heavy atoms of the proteins and phosphor atom of the lipids. The hydrogen-heavy atom bonds were constrained using the SHAKE algorithm and the simulation time step was set to 2.0 fs. The particle mesh Ewald (PME) method[85] was employed to compute the long-range electrostatic interactions and a cutoff value of 9.0 Å was applied to treat the non-bonded atomic interactions. The temperature was controlled using the Langevin thermostat with a collision frequency of $1.0\,ps^{-1}$. Each system was equilibrated using the constant number, volume, and temperature (NVT) ensemble at 310 K for 250 ps and under the constant number, pressure, and temperature (NPT) ensemble at 310 K and 1 bar for another 1 ns with constraints on the heavy atoms of the protein, followed by 10 ns short conventional MD (cMD) without any constraint.

The GaMD module implemented in the GPU version of AMBER18[80,81,86] was then applied to simulate the ACh-VU6007678-bound $M_5$ mAChR-mGα$_q$ and iperoxo-bound $M_5$ mAChR-mGα$_q$ complexes. The GaMD simulations included an 8-ns short cMD run to collect the potential statistics for calculating GaMD acceleration parameters, followed by a 56-ns GaMD equilibration after adding the boost potential. Finally, three independent 500-ns GaMD production simulations were conducted for each system with randomized initial atomic velocities. The average and standard deviation (SD) of the system potential energies were calculated every 800,000 steps (1.6 ns). All GaMD simulations were performed at the "dual-boost" level, where the reference energy was set to the lower bound. One boost potential was applied to the dihedral energetic term and the other to the total potential energetic term. The upper limit of the boost potential SD ($σ_0$) was set to 6.0 kcal/mol for both the dihedral and the total potential energetic terms.

For each system, the three GaMD production trajectories were combined for analysis. The CPPTRAJ software tool[87] was applied to calculate the time-courses of the root-mean-square derivations (RMSDs) of agonists and PAMs relative to the simulation starting structure, as well as the distances between VU6007678 and key interacting residues in the receptor, including F130$^{3.52}$, R146$^{4.41}$, M150$^{4.45}$, F126$^{3.48}$, R134$^{3.56}$, V123$^{3.45}$, T133$^{3.55}$ and K141$^{ICL2}$.

### Use of AI-assisted writing
The AI tool Claude 5 Sonnet (Anthropic, 2025) was used for proof-reading of the manuscript, checking for consistency in spelling, grammar, and clarity of the text. It was not used for generating ideas, content, figures, or data.

### Reporting summary
Further information on research design is available in the Nature Portfolio Reporting Summary linked to this article.

### Data availability
Atomic coordinates were deposited in the Protein Data Bank (PDB) under accession codes 9EK0 (iperoxo-bound $M_5$ AChR-mG$_q$-scFV16); 9EJZ (ACh-VU6007678-bound $M_5$ AChR-mG$_q$-scFv16-Nb35 complex).

Cryo-EM maps were deposited in the Electron Microscopy Data Bank under the accession codes EMD-48111 (consensus map for iperoxo-bound $M_5$ AChR-mG$_q$-scFV16); EMD-48109 (receptor focus map for iperoxo-bound $M_5$ AChR-mG$_q$-scFV16); EMD-48110 (consensus map for ACh-VU6007678-bound $M_5$ AChR-mG$_q$-scFv16-Nb35 complex); EMD-48108 (receptor focus map for ACh-VU6007678-bound $M_5$ AChR-mG$_q$-scFv16-Nb35 complex). Atomic coordinates for previously determined structures can be accessed via accession codes: 6OIJ, 6OIK,8E9Z, 7TRK, 7X2F, 6N48,5TZY, 7CFN, 6C1R, 6C1Q.

### Code availability
Initial coordinate, simulation input files, and combined imaged trajectory files for GaMD simulations of the ACh-VU6007678-bound, iperoxo-ML380-bound and ACh-bound $M_5$ mAChR-mGα$_q$ complexes in the public repository Figshare. The following are the links: [https://doi.org/10.6084/m9.figshare.28673348], [https://doi.org/10.6084/m9.figshare.28673351], [https://doi.org/10.6084/m9.figshare.29162954]. Source data are provided with this paper.

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

## Acknowledgements
This work was funded by the Australian Research council (ARC) discovery Project DP190102950 (CV, AC) and a National Health and Medical Research Council of Australia (NHMRC) Programme Grant APP1150083 (AC), NHMRC Project Grant APP1138448 (DMT), an NHMRC early career investigator Grant APP1196951 (DMT), a Discovery Early Career Researcher Award DE170100152 (DMT), and a US National Institutes of Health Grant R01GM132572 (YM). PRG was a Sir Keith Murdoch Fellow of the American Australian Association. CWL was funded by the William K. Warren Foundation. Cryo-EM imaging and sample vitrification were carried out at the Monash University Ramaciotti Centre for Cryo-Electron Microscopy. Data processing and storage of the cryo-EM datasets were supported by the Monash University MASSIVE high-performance computing facility and its supercomputing resources. This work used supercomputing resources with allocation awards TG-MCB180049 and BIO210039 through the US NSF-funded Advanced Cyberinfrastructure Coordination Ecosystem: Services & Support (ACCESS) programme and project M2874 through the US DOE National Energy Research Scientific Computing Center (NERSC).

## Author contributions
D.M.T. designed the overall research. W.A.C.B., J.I.M and D.M.T. designed, expressed, and purified protein samples. J.I.M, H.V. performed sample vitrification and cryo-EM imaging. W.A.C.B., J.I.M., D.M.T. processed the E.M. data and generated and analysed atomic models. JW., K.J., and Y.M. designed, performed, and analysed MD simulations. W.A.C.B., B.R, P.R.G. and M.Y. generated DNA constructs and performed pharmacology experiments. W.A.C.B., B.R, P.R.G., M.Y., A.C., C.V., and D.M.T. analysed pharmacology data. A.M.B. and C.W.L. provided chemical tools. Y.M., A.C., C.V., and D.M.T. provided supervision. W.AC.B., J.I.M. and D.M.T. wrote the manuscript with contributions and input from all authors.

## Competing interests
AC is a co-founder and holds equity in Septerna Inc. The remaining authors declare no competing interests.
