## [Transparent Peer Review file · Nature Communications]

Cryo-EM reveals an extrahelical allosteric binding site at the M5 mAChR

Corresponding Author: Dr David Thal

Version 0:

Reviewer comments:

Reviewer #1

(Remarks to the Author)

This study by Thal and colleagues presents the structural characterization of the M5 mAChR bound to an orthosteric agonist in the presence and absence of PAMs. mAChRs serve as model GPCRs in both physiological and pharmacological research. The recent success in developing selective M4 mAChR activators has renewed enthusiasm for designing subtype-selective PAMs and NAMs for this family of GPCRs. In this paper, the authors describe their efforts to identify a novel allosteric site located above ICL2 between TM3 and TM4 of the M5 mAChR for VU6007678, which is strongly supported by the high resolution cryo-EM map of the complex with VU6007678. Additional mutagenesis and MD simulations studies further confirm that VU6007678 binds to this site rather than previously characterized allosteric pockets in other mAChR subtypes. These findings provide new opportunities for structure-based drug design targeting M5 mAChR and other members of the mAChR family.

A major concern, which does not diminish the impact of this paper, is the insufficient discussion of the mechanism underlying the action of the PAM. The mutagenesis studies suggest that several residues in ICL2 and TM3/TM4 play important roles in the activity of VU6007678. However, the potential mechanisms by which VU6007678 functions as an ago-PAM for the receptor and achieves cooperativity with ACh through interactions with surrounding residues is not well explored.

The space above ICL2 surrounded by TM3, TM4, and TM5 is a common regulatory region in class A GPCRs and has been implicated in the action of allosteric modulators for other GPCRs, including NAMs for C5aR and PAMs for FFA1, the β 2-adrenergic receptor, and the D1 dopamine receptor. What drives the positive action of VU6007678? Why does it act as a PAM rather than a NAM? Does it stabilize a specific ICL2 conformation (<https://doi.org/10.1038/s41467-024-54103-6>), or does it influence the DRY motif? Additional simulations of the receptor with and without VU6007678 could provide valuable insights into its mechanism of action. Also, additional discussion of structural comparison with other GPCRs that have allosteric modulators bound in similar sites would be highly informative.

In line with this, Line 276, the authors stated that R146M eliminated allosteric agonism. However, Supplementary Figure 6 suggests that VU6007678 still functions as an ago-PAM, or allosteric agonist, at the R146M mutant (agonist activity independent of ACh), but the cooperativity with ACh is lost with this mutation. On the other hand, VU6007678 seems to increase both the potency and efficacy of ACh for the K141A mutant, whereas in the wt M5 receptor, it primarily increases potency without a significant effect on efficacy. The underlying mechanism behind these differential effects remains unclear and would be highly intriguing to explore, but it is not addressed in the study.

Reviewer #2

(Remarks to the Author)

M5 mAChR activation holds potential benefits for neurological disorders. In this study, the authors determine high-resolution cryo-EM structures of M5 mAChR bound to the agonist iperoxo, as well as co-bound with acetylcholine and the selective positive allosteric modulator (PAM) VU6007678. These findings complete the active-state structural characterization of the mAChR family and reveal a novel allosteric binding site on M5 mAChR, providing valuable insights into the mechanisms of allosteric regulation in mAChRs. The mutational functional analysis, combined with pharmacological assays and molecular

dynamics simulations, supports the key features of these structures.

Overall, the manuscript is well-organized, and the analyses are precise. However, there are several issues that the authors should clarify prior to publication.

1. Authors do not provide expression data for all mutants, are there any relationship between expression level and the functional modulation? particularly for F1303.52M, R1464.41M and M5/M2 swap mutants
2. Which Gq chimera was used to form a complex with the M5 receptor? The descriptions and labels for the Gq chimera are inconsistent: mini-G α qiN (main text), miniGs/q (Figure) or G mGsQi (Figure legend).
3. Does VU6007678 interact with G proteins, or if is it also contributes to the G protein selectivity for M5?

Additionally, there are some minor issues:

1. The panel labels in the figure should use lowercase letters and be consistent with main text.
2. The first paragraph referred the Table 1, I didn't find this table in main text manuscript.
3. Label the ligand name in supplementary Fig. 3e-f.
4. By contrast to the statement "Individual residue analysis revealed that the R1464.41M mutation reduced functional modulation and eliminated allosteric agonism...", the authors also described in the same paragraph that "Other single mutations (... , R1464.41M) showed no significant changes in affinity, functional modulation, or allosteric agonism". Based on provided results, the authors should revise the later sentence for consistency.
5. For supplementary Fig 2, please check if the values label of the y-axis is correct? Please also include the scale bar in the micrograph and 2D classification.

Reviewer #3

(Remarks to the Author)

In this paper, Burger et al. employed multiple approaches to investigate the new allosteric binding site of the M5 mAChR. However, this limitation should be addressed further within the text. The manuscript can be accepted after major revisions are made.

Major points:

The titles in the Results section are not appropriate. It would be more effective to use key findings or main points as titles to enhance clarity and readability.

The narrative logic of the Results section is unclear. The structure determination should be as the first part, and the title should be changed to "Structure determination of M5 mAChR in active state", rather than "Cryo-electron microscopy structure determination".

The descriptions should be concise. For instance, in the Materials and Methods section, it would be preferable to merge the "Gaussian Accelerated Molecular Dynamics" and "GaMD Simulations and Simulation Analysis" sections into a single section. This applies to other sections as well, where appropriate, to ensure clarity and brevity.

Lines 521-531. "Model building and redinement" is unsuitable for putting in front of "Cell culture". The descriptions of methods need to be coherent.

Line 651. Why not use the AMBER FF19SB force field?

Conclusion largely restates the results. Suggest refining to summarise key points and instead focus on significance of the findings and future directions.

Minor points:

Line 107, 679. Abbreviations should be defined at first mention and used consistently thereafter.

Line 114, 289. Table 1 and 4 are missing.

Lines 152 to 153. The value needs to be added to the figure.

Line 158. The writing style of Supplementary Fig. 4D is inconsistent with the others.

The relevant references should be cited, including but not limited to PDB:7TRP and 7MBY.

The reference list is incomplete, with some references including DOI links while others lacking them, and page numbers missing in reference 31.

Version 1:

Reviewer comments:

Reviewer #1

(Remarks to the Author)

The authors have addressed all of my concerns. I have only one suggestion, which is to add a reference PMID: 29867214 at line 413 when discussing NAMs at the extrahelical interface between TM3 and TM4 that inhibit the helical movements required for receptor activation. This paper provides structural evidence of two NAMs binding at the TM3/TM4 site.

Reviewer #2

(Remarks to the Author)

The authors have carefully addressed my concerns.

Reviewer #3

(Remarks to the Author)

The authors have thoroughly addressed all my comments raised in the previous review. The manuscript now presents a cohesive narrative with strengthened methodology validation. I recommend acceptance in its current form.

Response to Reviewers

Reviewer #1 (Remarks to the Author):

This study by Thal and colleagues presents the structural characterization of the M5 mAChR bound to an orthosteric agonist in the presence and absence of PAMs. mAChRs serve as model GPCRs in both physiological and pharmacological research. The recent success in developing selective M4 mAChR activators has renewed enthusiasm for designing subtype-selective PAMs and NAMs for this family of GPCRs. In this paper, the authors describe their efforts to identify a novel allosteric site located above ICL2 between TM3 and TM4 of the M5 mAChR for VU6007678, which is strongly supported by the high resolution cryo-EM map of the complex with VU6007678. Additional mutagenesis and MD simulations studies further confirm that VU6007678 binds to this site rather than previously characterized allosteric pockets in other mAChR subtypes. These findings provide new opportunities for structure-based drug design targeting M5 mAChR and other members of the mAChR family.

We appreciate the reviewer's kind remarks on our manuscript and their time in providing both positive and constructive feedback. We hope we have sufficiently addressed their concerns.

A major concern, which does not diminish the impact of this paper, is the insufficient discussion of the mechanism underlying the action of the PAM. The mutagenesis studies suggest that several residues in ICL2 and TM3/TM4 play important roles in the activity of VU6007678. However, the potential mechanisms by which VU6007678 functions as an ago-PAM for the receptor and achieves cooperativity with ACh through interactions with surrounding residues is not well explored.

The space above ICL2 surrounded by TM3, TM4, and TM5 is a common regulatory region in class A GPCRs and has been implicated in the action of allosteric modulators for other GPCRs, including NAMs for C5aR and PAMs for FFA1, the β 2-adrenergic receptor, and the D1 dopamine receptor. What drives the positive action of VU6007678? Why does it act as a PAM rather than a NAM? Does it stabilize a specific ICL2 conformation (<https://doi.org/10.1038/s41467-024-54103-6>), or does it influence the DRY motif? Additional simulations of the receptor with and without VU6007678 could provide valuable insights into its mechanism of action. Also, additional discussion of structural comparison with other GPCRs that have allosteric modulators bound in similar sites would be highly informative.

To further elaborate on the mechanism of action of VU6007678, we have expanded our discussion on the differences in the allosteric binding site between (1) the inactive and active states of M₅ mAChR and (2) the PAM-bound and unbound structures on lines 246-258. This section now provides additional insight into why VU6007678 functions as a PAM rather than a NAM and how it stabilizes the active state.

As suggested by the reviewer, we have added a new simulation of the M₅ mAChR without the PAM. In comparison, binding of the VU6007678 PAM induced new interactions with the receptor residues F3.52, R4.41, M4.45, R3.56, and Y3.51, formation of one more helical turn in ICL2, and reduced fluctuations in the ICL2 residues K141 and R142, as described on lines 260-265 and 325-327.

We have also added additional discussion on how VU6007678 acts as an ago-PAM, and the residues through which it achieves this, on lines 311-316.

We introduced a section in the discussion that compares other NAMs and PAMs that bind to the same site at other GPCRs, and their mechanism of action on lines 377-421.

In line with this, Line 276, the authors stated that R146M eliminated allosteric agonism. However, Supplementary Figure 6 suggests that VU6007678 still functions as an ago-PAM, or allosteric agonist, at the R146M mutant (agonist activity independent of ACh), but the cooperativity with ACh is lost with this mutation. On the other hand, VU6007678 seems to increase both the potency and efficacy of ACh for the K141A mutant, whereas in the wt M5 receptor, it primarily increases potency without a significant effect on efficacy. The underlying mechanism behind these differential effects remains unclear and would be highly intriguing to explore, but it is not addressed in the study.

We thank the reviewer for highlighting the discussion surrounding R146^{4.41}M. The data in Supplemental Figure 6 suggests a reduction in allosteric agonism of VU6007678 at R146^{4.41}M. However, as the reviewer correctly pointed out, VU6007678 can still function as an ago-PAM. The LogTau values (the operational efficacy parameter describing the degree of allosteric agonism) we have presented in Figure 5f and Supplementary Table 4 are corrected for expression levels that are also provided in Supplementary Table 4. R146^{4.41}M displays increased expression, consequently leading to an even more pronounced reduction in allosteric agonism when corrected for expression. To ensure accuracy, we have revised our phrasing on line 287 to clarify that, at R146^{4.41}M, there is a reduction rather than a complete elimination of allosteric agonism, as the data in Supplemental Figure 6 do not support the latter interpretation. This indicates that both the allosteric agonism and functional cooperativity of VU6007678 are similarly impacted at R146^{4.41}M.

Regarding the K141A mutant, we have added speculation that the increase in E_{max} could be related to differences in the conformational dynamics of ICL2, which is important for G protein signaling and selectivity. We note this will need to be studied more rigorously in future studies (lines 405-411).

Reviewer #2 (Remarks to the Author):

M5 mAChR activation holds potential benefits for neurological disorders. In this study, the authors determine high-resolution cryo-EM structures of M5 mAChR bound to the agonist iperoxo, as well as co-bound with acetylcholine and the selective positive allosteric modulator (PAM) VU6007678. These findings complete the active-state structural characterization of the mAChR family and reveal a novel allosteric binding site on M5 mAChR, providing valuable insights into the mechanisms of allosteric regulation in mAChRs. The mutational functional analysis, combined with pharmacological assays and molecular dynamics simulations, supports the key features of these structures.

Overall, the manuscript is well-organized, and the analyses are precise. However, there are several issues that the authors should clarify prior to publication.

We appreciate the reviewer's enthusiastic comments and for their help in identifying points that require clarification.

1. Authors do not provide expression data for all mutants, are there any relationship between expression level and the functional modulation? particularly for F1303.52M, R1464.41M and M5/M2 swap mutants

Expression data for mutants is found in Supplementary Table 3 and 4 under the column labelled 'Sites per Cell'. For F130^{3.52}M, R146^{4.41}M and M₅/M₂ swap mutants, no significant differences in expression were found compared to WT. We have previously reported expression data for the alanine mutants presented in Figure 1B, Supplementary Figure 1 and Supplementary Table 1 in PMID: 34351123. We have added a statement on line 117 to indicate this.

2. Which Gq chimera was used to form a complex with the M5 receptor? The descriptions and labels for the Gq chimera are inconsistent: mini-GαqiN (main text), miniGs/q (Figure) or GmGsQi (Figure legend).

Thank you for picking up this discrepancy. We have described our Gq chimera as mini-Gα_{sqiN} on line 128 at its first mention in order to accurately describe the chimera protein. Here we also state it will be referred to as mGα_q thereafter. In all further instances in the manuscript (including figures) we now refer to it as mGα_q.

3. Does VU6007678 interact with G proteins, or if is it also contributes to the G protein selectivity for M5?

VU6007678 does not interact with G protein. The M₅ mAChR is selective for Gq, irrespective of the presence of VU6007678.

Additionally, there are some minor issues:

1. The panel labels in the figure should use lowercase letters and be consistent with main text.

Thank you for picking up this discrepancy. We have adjusted this in all figures and figure legends.

2. The first paragraph referred the Table1, I didn't find this table in main text manuscript.

We have adjusted this accordingly to 'Supplementary Table 1'. Line 113

3. Label the ligand name in supplementary Fig. 3e-f.

Done

4. By contrast to the statement "Individual residue analysis revealed that the R1464.41M mutation reduced functional modulation and eliminated allosteric agonism...", the authors also described in the same paragraph that "Other single mutations (... , R1464.41M) showed no significant changes in affinity, functional modulation, or allosteric agonism". Based on provided results, the authors should revise the later sentence for consistency.

Thank you for picking up this mistake, we have adjusted the latter statement on line 303 by removing R146^{4.41}M.

5. For supplementary Fig 2, please check if the values label of the y-axis is correct? Please also include the scale bar in the micrograph and 2D classification.

We have adjusted the y-axis so that it is consistent and added scale bars to both the micrographs and 2D classes.

Reviewer #3 (Remarks to the Author):

In this paper, Burger et al. employed multiple approaches to investigate the new allosteric binding site of the M5 mAChR. However, this limitation should be addressed further within the text. The manuscript can be accepted after major revisions are made.

We thank the reviewer for taking the time to thoroughly review our manuscript and for providing advice on how to improve the manuscript.

Major points:

The titles in the Results section are not appropriate. It would be more effective to use key findings or main points as titles to enhance clarity and readability.

We have adjusted subheadings where appropriate on line 122.

The narrative logic of the Results section is unclear. The structure determination should be as the first part, and the title should be changed to “Structure determination of M₅ mAChR in active state”, rather than “Cryo-electron microscopy structure determination”.

We thank the reviewer for the suggestion and have renamed the subtitle of this section to “Structure determination of the M₅ mAChR in an active state” on line 122.

Given the well-established presence of allosteric sites in the M₅ mAChR, our results section first presents a comprehensive mutagenesis analysis of these reported sites. The absence of positive data confirming ML380 binding to these sites led us to pursue an ML380-bound M₅ mAChR structure. We believe it is essential to report our mutagenesis findings as they provide valuable insights, particularly as we were unable to obtain a ML380 bound structure. Presenting this data first offers the most accurate narrative as it reflects the scientific process we followed.

The descriptions should be concise. For instance, in the Materials and Methods section, it would be preferable to merge the "Gaussian Accelerated Molecular Dynamics" and "GaMD Simulations and Simulation Analysis" sections into a single section. This applies to other sections as well, where appropriate, to ensure clarity and brevity.

As suggested by the reviewer, we have merged the "Gaussian Accelerated Molecular Dynamics" and "GaMD Simulations and Simulation Analysis" method sections into a single section “**Gaussian accelerated Molecular Dynamics (GaMD), Simulations and Simulation Analysis**” in the revised manuscript.

Lines 521-531. “Model building and redinement” is unsuitable for putting in front of “Cell culture”. The descriptions of methods need to be coherent.

Our methods are structured into distinct sections—structural biology, pharmacology, and molecular dynamics—with subsections presented in the order they are performed. In the structural biology section, model building and refinement are the final steps, while in the pharmacology section, cell culture is the first method conducted.

Line 651. Why not use the AMBER FF19SB force field?

Previous studies comparing the AMBER FF14SB and FF19SB force fields revealed only minor differences in the dynamic behaviors of simulated systems (1-3). FF14SB remains more widely used in simulation studies of GPCRs (4-6). AMBER FF14SB has been used in the current study, but we may use FF19SB more in the future, like in one of our other recent studies (7).

References:

1. Paul, S. & Biswas, P. Dimerization of Full-Length A β -42 Peptide: A Comparison of Different Force Fields and Water Models. *ChemPhysChem* 25, e202400502 (2024).

2. Abriata, L. A. & Dal Peraro, M. Assessment of transferable forcefields for protein simulations attests improved description of disordered states and secondary structure propensities, and hints at multi-protein systems as the next challenge for optimization. *Computational and Structural Biotechnology Journal* 19, 2626-2636 (2021).
3. Coppa, C., Bazzoli, A., Barkhordari, M. & Contini, A. Accelerated Molecular Dynamics for Peptide Folding: Benchmarking Different Combinations of Force Fields and Explicit Solvent Models. *Journal of Chemical Information and Modeling* 63, 3030-3042 (2023).
4. Su, M. et al. Structural basis of agonist specificity of α 1A-adrenergic receptor. *Nature Communications* 14, 4819, doi:10.1038/s41467-023-40524-2 (2023).
5. Casiraghi, M. et al. Structure and dynamics determine G protein coupling specificity at a class A GPCR. *Science Advances* 11, eadq3971 (2025).
6. Yokoi, S., Suno, R. & Mitsutake, A. Structural and Computational Insights into Dynamics and Intermediate States of Orexin 2 Receptor Signaling. *The Journal of Physical Chemistry B* 128, 6082-6096 (2024).
7. Do HN, Wang J, & Miao Y Deep Learning Dynamic Allostery of G-Protein-Coupled Receptors. *JACS Au*, 3(11):3165-3180 (2023).

Conclusion largely restates the results. Suggest refining to summarise key points and instead focus on significance of the findings and future directions.

In line with the points raised by Reviewer #1, we have added an additional and expanded section in the discussion that compares and contrasts the mechanism of action for other NAMs and PAMs that bind at a similar allosteric site at other GPCRs on lines 377-421.

Minor points:

Line 107, 679. Abbreviations should be defined at first mention and used consistently thereafter.

Done

Line 114, 289. Table 1 and 4 are missing.

We have corrected this to Supplementary Table 1 and 4, respectively.

Lines 152 to 153. The value needs to be added to the figure.

We have incorporated the Å value into Supplementary Figure 4.

Line 158. The writing style of Supplementary Fig. 4D is inconsistent with the others.

Changed.

The relevant references should be cited, including but not limited to PDB:7TRP and 7MBY.

Is the reviewer referring to a specific section in the manuscript where these studies should be cited? PDB:7TRP corresponds to Vuckovic et al., *eLife* (2023), which is already cited as reference 34 and appears multiple times throughout the manuscript. Similarly, PDB:7MBY corresponds to Mobbs et al., *PLoS Biology*, cited as reference 39 and also referenced multiple times."

The reference list is incomplete, with some references including DOI links while others lacking them, and page numbers missing in reference 31.

We have adjusted this accordingly.

Response to Reviewers

We thank all three reviewers for their time and effort in reviewing our manuscript.

Reviewer #1 (Remarks to the Author):

The authors have addressed all of my concerns. I have only one suggestion, which is to add a reference PMID: 29867214 at line 413 when discussing NAMs at the extrahelical interface between TM3 and TM4 that inhibit the helical movements required for receptor activation. This paper provides structural evidence of two NAMs binding at the TM3/TM4 site.

We have added the suggested reference to the manuscript.

Reviewer #2 (Remarks to the Author):

The authors have carefully addressed my concerns.

Reviewer #3 (Remarks to the Author):

The authors have thoroughly addressed all my comments raised in the previous review. The manuscript now presents a cohesive narrative with strengthened methodology validation. I recommend acceptance in its current form.